# From Isolated Conversations to Hierarchical Schemas: Dynamic Tree Memory Representation for LLMs

**Alireza Rezazadeh, Zichao Li, Wei Wei, Yujia Bao**
Center for Advanced AI, Accenture
{alireza.rezazadeh,zichao.li,wei.h.wei,yujia.bao}@accenture.com

## Abstract

Recent advancements in large language models have significantly improved their context windows, yet challenges in effective long-term memory management remain. We introduce **MemTree**, an algorithm that leverages a dynamic, tree-structured memory representation to optimize the organization, retrieval, and integration of information, akin to human cognitive schemas. MemTree organizes memory hierarchically, with each node encapsulating aggregated textual content, corresponding semantic embeddings, and varying abstraction levels across the tree's depths. Our algorithm dynamically adapts this memory structure by computing and comparing semantic embeddings of new and existing information to enrich the model's context-awareness. This approach allows MemTree to handle complex reasoning and extended interactions more effectively than traditional memory augmentation methods, which often rely on flat lookup tables. Evaluations on benchmarks for multi-turn dialogue understanding and document question answering show that MemTree significantly enhances performance in scenarios that demand structured memory management.

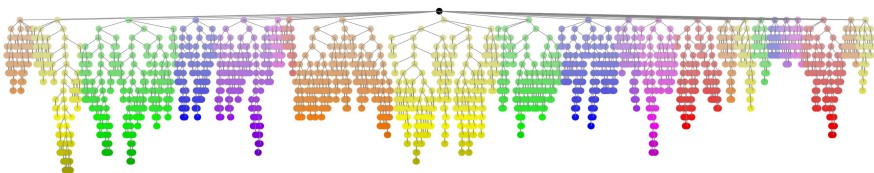

Figure 1: **MemTree** (subset) developed on the MultiHop dataset (Tang & Yang, 2024). MemTree updates its structured knowledge when new information arrives, enhancing inference-time reasoning capabilities of LLMs.

## 1 Introduction

Despite recent advances in large language models (LLMs) where the context window has expanded to millions of tokens (Ding et al., 2024; Bulatov et al., 2023; Beltagy et al., 2020; Chen et al., 2023; Tworkowski et al., 2024), these models continue to struggle with reasoning over long-term memory (Sun et al., 2021; Liu et al., 2024; Kuratov et al., 2024). This challenge arises because LLMs rely primarily on a key-value (KV) cache of past interactions, processed through a fixed number of transformer layers, which lack the capacity to effectively aggregate extensive historical data. Unlike LLMs, the human brain employs dynamic memory structures known as schemas, which facilitate the efficient organization, retrieval, and integration of information as new experiences occur (Anderson, 2005; Ghosh & Gilboa, 2014; Gilboa & Marlatte, 2017). This dynamic restructuring of memory is a cornerstone of human cognition, allowing for the flexible application of accumulated knowledge across varied contexts.

A prevalent method to address the limitations of long-term memory in LLMs involves the use of external memory. Weston et al. (2014) introduced the concept of utilizing external memory for the storage and retrieval of relevant information. More recent approaches in LLM research have

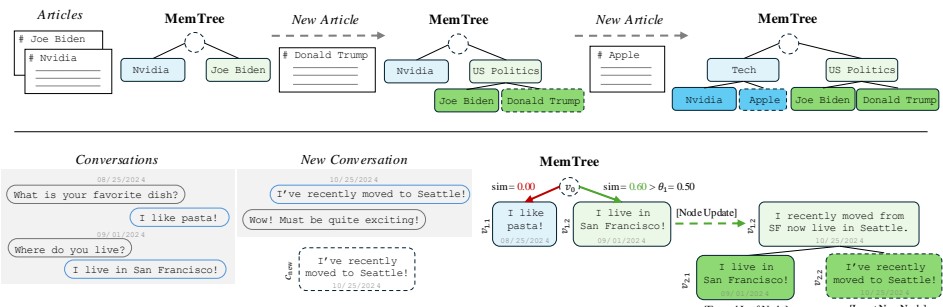

Figure 2: Illustration of MemTree. MemTree represents knowledge schema via a dynamic tree. Both parent and leaf nodes archive textual content, summarizing information relevant to their respective levels. Upon receiving new information, the system begins traversal from the root node. If the new information is semantically akin to an existing leaf node under the current node, it is routed to that node. Conversely, if it diverges from all existing leaf nodes under the current node, a new leaf node is created under the current node, concluding the traversal. During this process, all ancestor nodes will integrate the new information into the higher-level summaries they maintain.

explored various techniques to manage historical observations in databases, retrieving pertinent data for given queries through vector similarity searches in the embedding space (Park et al., 2023; Packer et al., 2023; Zhong et al., 2024). However, these methods primarily utilize a lookup table for memory representation, which fails to capture the inherent structure in data. Consequently, each past experience is stored as an isolated instance, lacking the interconnectedness and integrative capabilities of the human brain's schemas. This limitation becomes increasingly problematic as the size of the memory grows or when relevant information is distributed across multiple instances.

In this work, we introduce **MemTree**, an algorithm designed to emulate the schema-like structures of the human brain by maintaining a dynamically structured memory representation during interactions. Within MemTree, each memory unit is represented as a node within a tree, containing node-level information and links to child nodes.

Upon encountering new information, MemTree updates its memory structure starting from the root node. It evaluates at each node whether to instantiate a new child node or integrate the information into an existing child node. This decision process is governed by a traversal algorithm that efficiently adds new information with an insertion complexity of O(log N), where N denotes the number of conversational interactions. This structure facilitates the aggregation of knowledge at parent nodes, which evolve to capture high-level semantics as the tree expands. For knowledge retrieval, MemTree computes the cosine similarity between node embeddings and the query embedding. This method maintains the retrieval time complexity comparable to existing approaches, ensuring efficiency.

We evaluated MemTree across four benchmarks, covering both conversational and document question-answering tasks, and compared it against online and offline knowledge representation methods. MemTree enables seamless dynamic updates as new data becomes available, a capability characteristic of online methods. In contrast, offline methods require complete dataset access and are costly to update, as they involve periodic rebuilding to incorporate new information.

- In extended conversations, MemTree consistently outperforms other online methods, including MemoryStream (Park et al., 2023) and MemGPT (Packer et al., 2023), maintaining superior accuracy as discussions progress.

- For document question-answering, MemTree excels in two key areas. On single-document tasks, it outperforms other online methods and reduces the performance gap with offline models, particularly surpassing RAPTOR (Sarthi et al., 2024) on challenging questions that require deeper reasoning. In multi-document tasks, MemTree not only surpasses online methods but also approaches the performance of offline models, particularly outperforming GraphRAG (Edge et al., 2024). Moreover, on complex temporal queries requiring the analysis of event sequences across multiple documents, MemTree exceeds all offline methods.

These results demonstrate that MemTree is a robust and scalable solution, delivering high accuracy across diverse and challenging tasks while retaining the efficiency of online systems.

## 2 RELATED WORK

**Memory-Augmented LLMs**   Recent advancements in memory-augmented LLMs have introduced various strategies for enhancing memory capabilities. Park et al. (2023) developed LLM-based agents that log experiences as timestamped descriptions, retrieving memories based on recency, importance, and relevance. Similarly, Cheng et al. (2023) developed Selfmem with a dedicated memory selector. MemGPT (Packer et al., 2023) proposed automatic memory management through LLM function-calling for conversational agents and document analysis, providing a pre-prompt with detailed instructions on memory hierarchy and utilities, along with a schema for accessing and modifying the memory database. Zhong et al. (2024) introduced MemoryBank, a long-term memory framework that stores timestamped dialogues and uses exponential decay to forget outdated information. Additionally, Mitchell et al. (2022) proposed storing model edits in an explicit memory and learning to reason over them to adjust the base model's predictions. These methods represent common solutions for adding memory to LLMs, focusing on tabular memory storage and vector similarity retrieval (Zhang et al., 2024). However, as memory scales or information becomes dispersed across multiple entries, their unstructured representations reveal significant limitations.

Another line of work explores triplet-based memory. For example, Modarressi et al. (2023) proposed encoding relationships in triplets, and Anokhin et al. (2024) extended this approach to graph-based triplet memory for text-based games. While effective at encoding individual relations or scene graphs at the object level, these methods struggle with scalability and generalization to more complex data that do not fit neatly into a strict triplet format.

**Structured Retrieval-Augmented Generation Approaches**   To address the limitations of unstructured memory representations, recent advances have integrated structured knowledge into RAG models, enhancing navigation and summarization in complex QA tasks. Trajanoska et al. (2023) leveraged LLMs to extract entities and relationships from unstructured text to construct knowledge graphs. Similarly, Yao et al. (2023) proposed techniques to fill in missing links and nodes in existing knowledge graphs by utilizing LLMs to infer unseen relationships. Ban et al. (2023) identified causal relationships within textual data and represented them in graph form to enhance understanding of causal structures. In the context of retrieval-augmented generation, Gao et al. (2023) provided a comprehensive review of existing RAG methods, and Baek et al. (2023) utilized knowledge graphs as indexes within RAG frameworks for efficient retrieval of structured information.

More relevant to our work, RAPTOR (Sarthi et al., 2024) organizes text into a recursive tree, clustering and summarizing chunks at multiple layers to enable efficient retrieval of both high-level themes and detailed information. GraphRAG (Edge et al., 2024) constructs a knowledge graph from LLM-extracted entities and relations, partitioning it into modular communities that are independently summarized and combined via a map-reduce framework. While these methods effectively structure large textual data to improve retrieval and generation capabilities, they are limited to static corpora, requiring full reconstruction to integrate new information, and do not support online memory updates.

In this work, we propose a novel structured memory representation that overcomes these limitations by enabling dynamic updates and efficient retrieval in large-scale memory systems without necessitating full reconstruction.

## 3 METHOD

MemTree represents memory as a tree $T = (V, E)$, where $V$ is the set of nodes, and $E \subseteq V \times V$ is the set of directed edges representing parent-child relationships. Each node $v \in V$ is represented as:

$$v = [c_v, e_v, p_v, \mathcal{C}_v, d_v]$$

where:

- $c_v$: the textual content aggregated at node $v$.
- $e_v \in \mathbb{R}^d$: an embedding vector derived using an embedding model $f_{\text{emb}}(c_v)$.
- $p_v \in V$: the parent of node $v$.
- $\mathcal{C}_v \subseteq V$: the set of children of node $v$, with edges directed from $v$ to each $u \in \mathcal{C}_v$.

- $d_v$: the depth of node $v$ from the root node $v_0$.

Note that the root node $v_0$ serves as a structural node, containing neither content nor embedding, i.e., $c_{v_0} = \varnothing$ and $e_{v_0} = \varnothing$.

MemTree utilizes this tree-structured representation to dynamically track and update the knowledge exchanged between the user and the LLM. While less flexible than a generic graph architecture, the tree structure inherently biases the model towards hierarchical representation. Additionally, trees offer efficient complexity for insertion and traversal, making the algorithm suitable for real-time online use cases.

When new information is observed, MemTree dynamically adapts by traversing the existing structure, identifying the appropriate subtree for integration, and updating relevant nodes (Section 3.1). This process, illustrated in Figure 2, ensures the proper integration of new information while preserving the underlying context and hierarchical relationships within the memory. When retrieving information from the memory, MemTree simply compares the embeddings of the query message with the embeddings of each node in the tree, returning the most relevant nodes (Section 3.2).

### 3.1 MEMORY UPDATE

The memory update procedure in MemTree is triggered upon observing new information (e.g., a new conversation). This procedure ensures that the tree structure dynamically adapts and integrates new data while maintaining a coherent hierarchical representation. The complete memory update process is outlined in Algorithm 1.

**Attaching New Information by Traversing the Existing Tree**    To integrate new information, we begin by creating a new node $v_{\text{new}}$ with the textual content $c_{v_{\text{new}}}$. Then we start tree traversal from the root node. At each node $v$, MemTree evaluates the *semantic similarity* between the new information $c_{v_{new}}$ and the children of the current node in the embedding space. This evaluation is performed by computing the embedding $e_{v_{\text{new}}} = f_{\text{emb}}(c_{v_{\text{new}}})$ for the new content $c_{v_{\text{new}}}$ and comparing it to the embeddings of the child nodes $\mathcal{C}(v)$ of the current node $v$ using cosine similarity.

This similarity evaluation drives the following decisions:

- **Traverse Deeper:** If a child node's similarity exceeds a depth-adaptive threshold $\theta(d_v)$, traversal continues along that path. If multiple child nodes meet this criterion, the path with the highest similarity score is chosen.
    - **Boundary:** When traversal reaches a leaf node, the leaf is expanded to become a parent node, accommodating both the original leaf node and $v_{\text{new}}$ as children. The parent's content is then updated to aggregate both the original leaf node's content and the new information $c_{v_{\text{new}}}$. The details of this aggregation process will be explained below.
- **Create New Leaf Node:** If all child nodes' similarities are below the threshold $\theta(d_v)$, $v_{\text{new}}$ is directly attached as a new leaf node under the current node.

The similarity threshold $\theta(d)$ is adaptive based on the node's depth $d$, defined as:

$$\theta(d) = \theta_0 e^{\lambda d},$$

where $\theta_0$ is the base threshold, and $\lambda$ controls the rate of increase with depth. This mechanism ensures that deeper nodes, which represent more specific information, require a higher similarity for new data integration, thereby preserving the tree's hierarchical integrity. Further details, including specific parameter values, are provided in the Appendix A.1.3.

**Updating Parent Nodes Along the Traversal Path**    Once $v_{\text{new}}$ is inserted, the content and embeddings of all parent nodes $v$ along the traversal path are updated to reflect the new information. This is achieved through a conditional aggregation function:

$$c'_v \leftarrow \text{Aggregate}(c_v, c_{\text{new}} \mid n),$$

where $c'_v$ is the updated content, and $n = |\mathcal{C}(v)|$ is the number of descendants of node $v$. The aggregation function, implemented as an LLM-based operation, combines the existing content $c_v$

with the new content $c_{\text{new}}$, conditioned on $n$. As $n$ increases, the aggregation abstracts the content further to balance the existing and new information (see Appendix A.1.2 for more details).

The embedding of the parent node is then updated as:

$$e_v \leftarrow f_{\text{emb}}(c'_v),$$

ensuring that the parent node effectively represents both the new and existing information. This process maintains the hierarchical organization of the memory as the tree expands, enabling MemTree to adaptively and accurately represent the evolving conversation. A key advantage of this method is its computational efficiency: once the traversal path is defined, the content aggregation and embedding updates for parent nodes can be parallelized on the CPU. This significantly accelerates the update process, reducing bottlenecks as the memory grows.

**Connection to Online Hierarchical Clustering Algorithms**  Our memory update algorithm can be viewed as an instance of online hierarchical clustering algorithms (Zhang et al., 1996; Kobren et al., 2017). We draw inspiration from the OTD (Online Top-Down Clustering) algorithm proposed by Menon et al. (2019), which enables efficient online updates by calculating inter- and intra-subtree similarities during the insertion process. This algorithm is known to provably approximate the Moseley-Wang revenue (Moseley & Wang, 2017). In this work, we relax the inter- and intra-subtree similarity comparisons by utilizing semantic similarities in the embedding space. We achieve this by formatting the subtree representation (parent nodes) with LLMs as we traverse the MemTree during the memory update.

**Theorem 1** (Approximation Guarantee of MemTree (Informal)). Assuming the data processed by MemTree satisfies the $\beta$-well-separated condition (see Appendix B.4), the hierarchy maintained by MemTree achieves a revenue

$$\text{Rev}(\text{MemTree}; W) \geq \beta/3 \cdot \text{Rev}(T^*; W),$$

where $T^*$ is the optimal hierarchy maximizing the Moseley-Wang revenue.

This alignment with OTD ensures a high-quality hierarchical memory representation in MemTree. Further details and proofs are provided in Appendix B.

## 3.2 MEMORY RETRIEVAL

Efficient and effective retrieval of relevant information is crucial for ensuring that MemTree can provide meaningful responses based on past conversations. Inspired by RAPTOR (Sarthi et al., 2024), we adopt the collapsed tree retrieval method, which offers significant advantages over traditional tree traversal-based retrieval.

**Collapsed Tree Retrieval**  The collapsed tree approach enhances the search process by treating all nodes in the tree as a single set. Instead of conducting a sequential, layer-by-layer traversal, this method flattens the hierarchical structure, allowing for simultaneous comparison of all nodes. This technique simplifies the retrieval process and ensures a more efficient search.

The retrieval process involves the following steps:

1. Query Embedding: Embed the query $q$ using $f_{\text{emb}}(q)$ to obtain $e_q$.
2. Similarity Computation: Calculate cosine similarities between $e_q$ and all tree nodes.
3. Filtering: Exclude nodes with similarity scores below a threshold $\theta_{\text{retrieve}}$.
4. Top-K Selection: Sort the remaining nodes by similarity and select the top-k most relevant nodes.

## 4 EXPERIMENTS

### 4.1 DATASETS

We evaluate the effectiveness of MemTree across various settings using four datasets: Multi-Session Chat, Multi-Session Chat Extended, QuALITY, MultiHop RAG. These datasets were selected to

represent different interactive contexts—dialogue interactions and information retrieval from multiple texts, respectively, providing a comprehensive test bed for our model. Additional statistics for each dataset can be found in Appendix A.2.1.

- Multi-Session Chat (**MSC**): The dataset was introduced by (Xu, 2021). In this work, we consider the processed version provided by (Packer et al., 2023). The dataset consists of 500 sessions, each featuring approximately 15 rounds of synthetic dialogue between two agents. Each session includes follow-up questions that challenge the model to retrieve and utilize information from prior dialogues within the same session. For each session, a memory representation is independently built, capturing the dialogue rounds as they unfold.

- Multi-Session Chat Extended (**MSC-E**): To test the performance for even longer conversation rounds, we expanded MSC by generating an additional 70 sessions, each containing about 200 rounds of dialogue. In these extended sessions, a follow-up question follows each conversation round, demanding more precise and timely information retrieval across the dialogues. As in MSC, memory representations are constructed independently for each session. We detail the extension methodology in Appendix A.2.3.

- Single-Document Question Answering (**QuALITY**): The QuALITY dataset, introduced by Pang et al. (2021), consists of context passages averaging 5,000 tokens, paired with multiple-choice questions that require reasoning across entire documents to answer. A memory representation is built independently for each document. The dataset is divided into two subsets: QuALITY-Easy and QuALITY-Hard. The latter contains questions that most human annotators found challenging to answer within the time constraints. While the original dataset was designed for multiple-choice question answering, in this paper we explore the more difficult setting where the model must generate the answer directly, without being provided with the four answer options.

- Multi-Document Question Answering (**MultiHop RAG**): This dataset comprises 609 distinct news articles across six categories (Tang & Yang, 2024). It includes 2,556 multi-hop questions requiring the integration of information from multiple articles to formulate comprehensive answers. We consider three question types: inference, comparison, and temporal reasoning, each adding a layer of complexity to the information retrieval process. All news articles are used to construct a unified memory representation, which is queried to answer the multi-hop questions.

## 4.2 BASELINES

We compare MemTree with various baseline methods along with a **naive baseline**, which involves concatenating all chat histories and feeding them into a large language model (LLM):

- **MemoryStream**: Park et al. (2023) proposes a flat lookup-table style memory that logs chat histories through an embedding table. The primary distinction between MemTree and this baseline is that MemTree utilizes a structured tree representation for the memory and models high-level representations throughout the memory insertion process.

- **MemGPT**: (Packer et al., 2023) introduces a memory system designed to update and retrieve information efficiently. It uses an OS paging algorithm to evict less relevant memory into external storage. However, like MemoryStream, it does not format high-level representations.[1]

- **RAPTOR**: Sarthi et al. (2024) constructs a structured knowledge base using hierarchical clustering over all available information. The key difference between MemTree and this baseline is that MemTree operates as an *online* algorithm, updating the tree memory representation on-the-fly based on incoming knowledge, while RAPTOR applies hierarchical clustering on a fixed dataset.[2].

- **GraphRAG**: Edge et al. (2024) introduces a graph-based indexing approach designed to improve query-focused summarization and extract global insights from large text corpora. Like RAPTOR, GraphRAG assumes access to the entire corpus and applies the Leiden algorithm to identify community structures within the document graph. However, while MemTree expands its memory top-down to allow for efficient, online updates, GraphRAG generates community summaries in a bottom-up fashion, which is less suited for real-time adaptability.[3]

---

[1]`https://github.com/cpacker/MemGPT`
[2]`https://github.com/parthsarthi03/raptor`
[3]`https://github.com/microsoft/graphrag`

Table 1: Naive History Combination vs. External Memory on MSC. With only 15 dialogue rounds (<1,000 tokens), concatenating the entire history to GPT-4o achieves the best performance. Among query-only models, MemTree outperforms MemGPT and MemoryStream in accuracy and ROUGE.

| Model | Context | Accuracy ⇑ | ROUGE-L (R) ⇑ |
|-------|---------|------------|---------------|
| *Results reported by (Packer et al., 2023)* | | | |
| GPT-4 Turbo | Query + Full history summary | 35 | 35 |
| GPT-4 Turbo | Query + Full history summary + MemGPT | 93 | 82 |
| *Our results with GPT-4o and text-embedding-3-large* | | | |
| GPT-4o | Query + Full history | **95.6** | **88.0** |
| GPT-4o | Query + MemGPT | 70.4 | 68.6 |
| GPT-4o | Query + MemoryStream | 84.4 | 79.1 |
| GPT-4o | Query + **MemTree** | 84.8 | 79.9 |

Table 2: **Accuracy** on MSC-E. The MSC-E dataset extends MSC from 15 to 200 dialogue rounds, providing a better test for long-context reasoning. Both MemoryStream and MemTree outperform the naive baseline, highlighting the importance of external memory. Overall accuracy and a breakdown by evidence position are shown; standard deviations are in Figure A.1.

| Model | Context | Position of the supporting evidence | | | | | Overall |
|-------|---------|------|-------|--------|---------|---------|---------|
| | | *0-40* | *40-80* | *80-120* | *120-160* | *160-200* | |
| GPT-4o | Query + Full history | **84.5** | 78.3 | 75.5 | 74.4 | 76.7 | 78.0 |
| GPT-4o | Query + MemoryStream | 78.5 | 81.0 | 81.0 | 81.4 | 81.8 | 80.7 |
| GPT-4o | Query + **MemTree** | 82.1 | **82.1** | **82.3** | **82.3** | **84.2** | **82.5** |

To demonstrate the applicability of our approach, we consider both open-source and commercial models in our experiments. For LLMs, we used OpenAI's `GPT-4o` (version 2024-05-13) and `Llama-3.1-70B-Instruct` (Dubey et al., 2024). For the embedding models, we employed `text-embedding-3-large` and `E5-Mistral-7B-Instruct` (Wang et al., 2023). In each experiment, we standardized the use of the LLM and embedding model across all baselines to ensure that any performance differences observed were attributable to the memory management methodologies, rather than variations in the models' capabilities or embeddings.

## 4.3 Implementation Details and Evaluation Metrics

Following previous work (Packer et al., 2023; Tang & Yang, 2024), we report the end-to-end question answering performance. Given each context-question-answer tuple, the experimental procedure involves four steps:

1. Load the corresponding dialogue/history into the memory.
2. Retrieve the relevant information from the memory based on the given query.
3. Use GPT-4o to answer the query based on the retrieved information.
4. Evaluate the generated answer using one of the following two metrics: 1) Use GPT-4o to compare the generated answer with the reference answer, resulting in a binary accuracy score; 2) Evaluate the ROUGE-L recall (R) metric of the generated answer compared to the relatively short gold answer labels, without involving the LLM judge.

The detailed prompts for steps 3 and 4 can be found in Appendix A.2. Other implementation details for MemTree can be found in Appendix A.1.

## 5 Results

### 5.1 Multi-Session Chat

**15-round dialogue** We present the MSC results in Table 1. For the naive baseline, directly providing the full history to GPT-4o yields the best result, achieving an accuracy of 96%. This outcome is expected, given that the entire dialogue consists of only 15 rounds and fewer than one thousand tokens. We also note that providing a summary of the chat history significantly drops performance

Table 3: **Accuracy** on QuALITY. Performance is evaluated on (1) Easy questions, answerable with surface-level information, and (2) Hard questions, requiring deeper reasoning. MemTree shows strong overall performance, surpassing online methods and nearing offline methods in both categories.

| Model | Context | Easy | Hard | Overall |
|-------|---------|------|------|---------|
| Llama-3.1-70B | Query + Full text | 70.1 | 60.3 | 65.1 |
| *Offline Method* | | | | |
| Llama-3.1-70B | Query + RAPTOR | 65.2 | 53.0 | 59.0 |
| Llama-3.1-70B | Query + GraphRAG | 65.9 | 59.8 | 62.8 |
| *Online Method* | | | | |
| Llama-3.1-70B | Query + MemoryStream | 46.7 | 41.0 | 43.8 |
| Llama-3.1-70B | Query + **MemTree** | **63.3** | **56.5** | **59.8** |

to 35%, even for the more powerful GPT-4 Turbo model (Packer et al., 2023). This decline occurs because the summary may not cover the topics the query is addressing. To directly compare the performance of different memory management algorithms, we consider the setting where only the query and the retrieved information are provided to the LLM. In this scenario, MemTree surpasses both MemStream and MemGPT.[4]

**200-round dialogue**   Table 2 presents the results on MSC-E. We observe that both MemoryStream and MemTree achieve better overall accuracy than the naive baseline, which directly uses the full history. This illustrates the importance of having an external memory system as the conversation history grows. When we break down the accuracies based on the positions of the supporting evidence within the entire dialogue, we find that the naive baseline performs best when the evidence is presented early on, likely due to position bias (Liu et al., 2024). It is worth noting that since MemTree updates the memory sequentially based on the order of the dialogue, it inherently favors more recent conversations over older ones. This bias is demonstrated in Table 2, where the accuracy increases from 82.1 to 84.2. Nevertheless, MemTree consistently outperforms MemoryStream across all positions (see Figure A.1 for a visualization).

## 5.2   SINGLE-DOCUMENT QUESTION ANSWERING

Table 3 presents the accuracy of various models on the QuALITY benchmark. Llama-3.1 70B, which processes the full text in a single pass, achieves the highest overall accuracy at 65.1%. This superior performance is attributed to the dataset's relatively short length (5000 tokens), a trend also observed with the MSC dataset. Offline RAG methods such as RAPTOR and GraphRAG, designed for handling knowledge retrieval over longer contexts, achieve lower accuracies of 59.0% and 62.8%, respectively. The current online memory update method, MemoryStream, struggles with efficiently extracting memory key-value pairs, resulting in a significantly lower accuracy of 43.8%. In contrast, our method, MemTree, matches the offline performance of RAPTOR with a slightly higher accuracy of 59.8%, especially excelling on hard questions that demand deeper reasoning and comprehension. Moreover, MemTree retains the advantage of being an online method, allowing for continuous memory updates at minimal computational cost. Refer to Figure A.2 for a visualization of the results.

## 5.3   MULTI-DOCUMENT QUESTION ANSWERING

Table 4 summarizes the end-to-end performance of MultiHop RAG using various memory retrieval algorithms. All methods perform exceptionally well on inference-style questions, which focus on fact-checking based on a single document, consistently achieving over 95% accuracy. However, when it comes to more complex questions—those requiring the comparison of multiple documents or temporal reasoning—MemTree significantly outperforms MemoryStream, achieving a 9.1 percentage point advantage. Moreover, despite RAPTOR having full access to all information, MemTree's overall performance is within just 0.5 percentage points of this offline method. See Figure A.3 for a detailed visualization of these results.

Another observation from the table is that while humans can annotate evidence fairly accurately for inference and comparison-style questions, the annotated evidence for temporal questions is less

---

[4]We were unable to reproduce the results with the existing MemGPT GitHub codebase.

Table 4: **Accuracy** on MultiHop RAG. Results are shown for (1) Inference queries, (2) Comparison queries, and (3) Temporal queries. MemTree outperforms MemoryStream on comparison and temporal queries, narrowing the gap to the offline RAPTOR.

| Model | Context | Inference | Comparison | Temporal | Overall |
|---|---|---|---|---|---|
| GPT-4o | Human Annotated Evidence | 98.4 | 80.1 | 55.6 | 79.2 |
| *Offline method* | | | | | |
| GPT-4o | Query + RAPTOR | 96.6 | 76.5 | 66.0 | 81.0 |
| GPT-4o | Query + GraphRAG | 96.0 | 69.8 | 66.3 | 78.3 |
| *Online method* | | | | | |
| GPT-4o | Query + MemoryStream | **96.1** | 64.8 | 59.3 | 74.7 |
| GPT-4o | Query + **MemTree** | 96.0 | **73.9** | **68.4** | **80.5** |

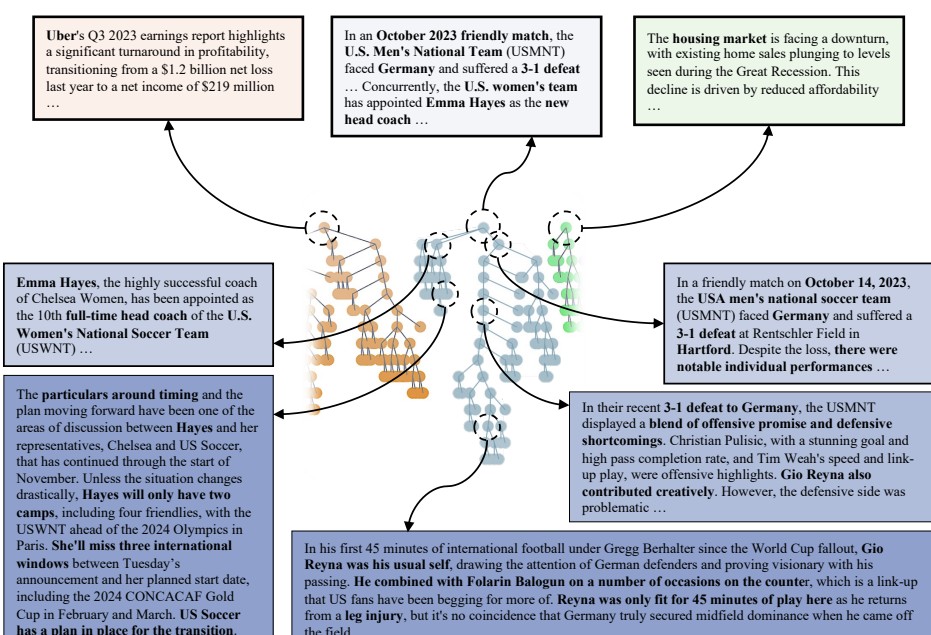

Figure 3: Visualization of the Learned MemTree Structure on the MultiHop RAG Dataset. Due to space limitations, we display only a small subtree from the entire tree (a larger subtree is depicted in Figure 1). As we traverse deeper into the tree, the content stored in the nodes becomes increasingly specific. For instance, the three blue nodes shown in the bottom right corner begin with a general summary of the *USMNT's 3-1 defeat to Germany*, then branch into *specific insights on individual performances and team dynamics*, and ultimately delve into *a detailed analysis of Gio Reyna's impact during the match*. Note that all intermediate contents in the parent nodes are generated by MemTree during the node update step. This hierarchical organization demonstrates how MemTree efficiently stores and retrieves information, progressing from overarching concepts to specific details.

precise. This results in worse performance than the model-derived memory for temporal questions. Importantly, MemTree excels on temporal reasoning tasks, surpassing all baselines, including offline approaches and human-annotated evidence.

**Statistics of the Learned MemTree**  Table 5 presents statistics for the learned MemTree on the Multihop RAG dataset, which consists of 609 documents. The resulting tree contains 3,154 nodes, with a maximum depth of 13 and an average branching factor of 2.1. Figure 4 illustrates the distribution of nodes across different depth levels, revealing that the majority of nodes are concentrated between depths 3 to 5. As the tree deepens, the information stored in the nodes increases in length. For instance, at depth 1 (just below the root), the median token count is slightly over 200, with a small deviation. By depth 10 and beyond, the median token count grows to around 800, with greater variability (see Figure 4).

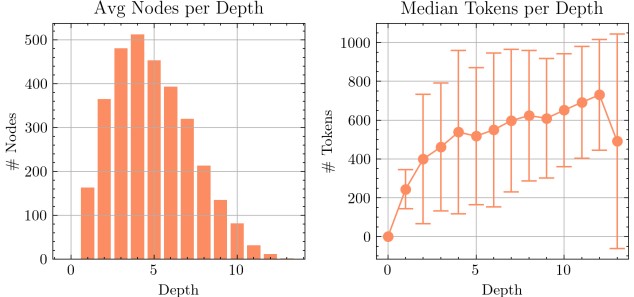

Figure 4: Depth-based Stats of MemTree learned on Multi-Hop RAG

Table 5: Overall Stats of MemTree learned on MultiHop

| MemTree Property | Value |
|---|---|
| #Nodes | 3164 |
| #Leaf Nodes | 1706 |
| #Branching Nodes | 1458 |
| Depth (max) | 13 |
| Depth (average) | 4.9 |
| Branching Factor | 2.1 |
| Height to Width Ratio | 6.5 |

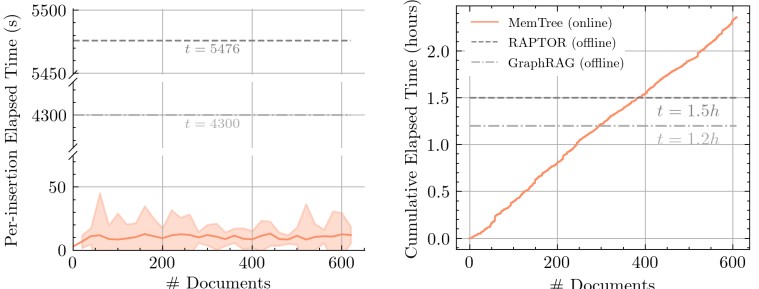

Figure 5: Efficiency of MemTree vs. RAPTOR and GraphRAG: MemTree's top-down insertion strategy allows content aggregation and embedding updates to be parallelized on the CPU, significantly accelerating memory updates as memory grows. Despite its cumulative cost being approximately 1.4x higher than the offline algorithms (RAPTOR and GraphRAG), it remains manageable. Results are reported on the MultiHop dataset.

**Hierarchical Representation of the MemTree**   The hierarchical structure of the learned MemTree reflects a semantic organization. Higher-level nodes capture more abstract, generalized information, while deeper nodes store finer details. Figures 1 and 3 further visualize this hierarchy. The model effectively groups related concepts, with intermediate parent nodes summarizing high-level information during memory insertion. This structure enables the MemTree to maintain a balance between abstract representations at the top and specific details at the bottom.

**Time Efficiency of Online Algorithm vs Offline Algorithm**   MemTree's continuous updates during conversations make it ideal for real-time scenarios. Once the traversal path is defined, its top-down insertion allows parent node updates to be parallelized on the CPU, accelerating the update process and reducing bottlenecks as memory grows. In contrast, RAPTOR and GraphRAG use clustering in a RAG setup, making memory updates after index construction impossible or costly. As shown in Figure 5, MemTree inserts new information in 10 seconds on average, while RAPTOR and GraphRAG take over an hour to build the full memory tree, making it impractical for real-time use. Although MemTree's cumulative time cost is 1.4x higher than RAPTOR's due to continuous updates, this trade-off enables maintaining an up-to-date memory in real time.

## 6   CONCLUSION

MemTree effectively addresses the long-term memory limitations of large language models by emulating the schema-like structures of the human brain through a dynamic tree-based memory representation. This approach enables efficient integration and retrieval of extensive historical data, as demonstrated by its superior performance on four benchmarks with different interactive contexts. Our evaluations reveal that MemTree consistently maintains high performance and demonstrates human-like knowledge aggregation by capturing the semantics of the context within its tree memory structure. This advancement offers a promising solution for enhancing the reasoning capabilities of LLMs in handling long-term memory.

**Ethical Statement**    In developing **MemTree**, we commit to ensuring that no private or proprietary data is mishandled during our experiments, and all data used for training and evaluation are publicly available. While our current research does not explicitly address principles such as transparency, responsibility, inclusivity, bias mitigation, or user safety, we recognize that recent advancements in these areas can be integrated into the memory learning component of our algorithm. We encourage the research community to engage with these ethical considerations as we strive to enhance our understanding and implementation of responsible AI practices.

**Reproducibility Statement**    We provide comprehensive details for reproducing our results in Section 4 and the Appendix, including our experimental setup, evaluation metrics, and implementation settings. The code and scripts used in our experiments will be made publicly available upon acceptance. All external libraries and dependencies required for reproduction are specified. Our method has been evaluated on both open-source and commercial models to demonstrate its applicability.

**Disclaimer**    This content is provided for general information purposes and is not intended to be used in place of consultation with our professional advisors. This document may refer to marks owned by third parties. All such third-party marks are the property of their respective owners. No sponsorship, endorsement or approval of this content by the owners of such marks is intended, expressed or implied.

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

## A  APPENDIX

### A.1  MEMTREE DETAILS

Further details and parameter settings for our approach are outlined below. Unless otherwise specified, these settings are consistent across all experiments presented in the paper.

#### A.1.1  MEMTREE ALGORITHM

The following outlines the algorithmic procedure for incrementally updating and restructuring the memory representation in MemTree. This approach ensures that new information is efficiently integrated into the existing memory hierarchy while dynamically adjusting based on content similarity and structural depth.

**Parameters:**

- $c$: the textual content stored at a node or introduced as new information.
- $e$: the embedding vector representing the content, generated by an embedding function $f_{\text{emb}}$.
- $v$: a node in the memory tree, which contains content, embeddings, and connections to other nodes. Note that the root is a structural node and does not hold content.
- $d$: the depth of a node in the tree.

---

**Algorithm 1** Adding New Information to MemTree

---

**Require:** New information $c_{\text{new}}$, root node $v_0$, threshold function $\theta(d)$

1: $e_{\text{new}} \leftarrow f_{\text{emb}}(c_{\text{new}})$
2: INSERTNODE($v_0, e_{\text{new}}, c_{\text{new}}, 0$)
3: **procedure** INSERTNODE($v, e_{\text{new}}, c_{\text{new}}, d$)
4:     **if** $v$ is a leaf **then**
5:         Expand $v$ into a parent
6:         Create and attach child node $v_{\text{leaf}}$ with original content
7:     **end if**
8:     Compute similarity $s_i = \text{sim}(e_{\text{new}}, e_i)$ for each child $v_i$ of $v$
9:     $v_{\text{best}} \leftarrow \arg\max(s_i), s_{\text{max}} \leftarrow \max(s_i)$
10:     **if** $s_{\text{max}} \geq \theta(d)$ **then**
11:         $c_v \leftarrow \text{Aggregate}(c_v, c_{\text{new}})$
12:         $e_v \leftarrow f_{\text{emb}}(c_v)$
13:         INSERTNODE($v_{\text{best}}, e_{\text{new}}, c_{\text{new}}, d+1$)
14:     **else**
15:         Create and attach new child node $v_{\text{child}}$ with $c_{\text{new}}$
16:     **end if**
17: **end procedure**

---

#### A.1.2  AGGREGATE OPERATION

When new information is added, the content of parent nodes along the traversal path is updated through a conditional aggregation. This process combines the existing content of the parent node with the new content, factoring in the number of its descendants. The aggregation operation is implemented using the following prompt:

```
You will receive two pieces of information:  New Information
is detailed, and Existing Information is a summary from
{n_children} previous entries.  Your task is to merge these
into a single, cohesive summary that highlights the most
important insights.
- Focus on the key points from both inputs.
- Ensure the final summary combines the insights from both
pieces of information.
```

```
    - If the number of previous entries in Existing Information
    is accumulating (more than 2), focus on summarizing more
    concisely, only capturing the overarching theme, and getting
    more abstract in your summary.
    Output the summary directly.
    [New Information]
    {new_content}
    [Existing Information (from {n_children} previous entries)]
    {current_content}
    [ Output Summary ]
```

### A.1.3 ADAPTIVE SIMILARITY THRESHOLD

The adaptive similarity threshold ensures that deeper nodes, representing more specific information, require higher similarity for new data integration, while shallower nodes are more abstract and accept broader content. This mechanism preserves the tree's hierarchical integrity by adjusting selectivity based on the node's depth. The threshold is computed as:

$$\text{threshold} = \text{base\_threshold} \times \exp\left(\frac{\text{rate} \times \text{current\_depth}}{\text{max\_depth}}\right)$$

where:

- $\text{base\_threshold} = 0.4$
- $\text{rate} = 0.5$
- current_depth is the depth of the current node.
- max_depth is the maximum depth of the tree.

### A.1.4 RETRIEVAL

For the MSC experiment, the retrieval system returns the top $k = 3$ similar dialogues from 15-round conversations, with a context length of 1000 tokens for all models. In the MSC-E dataset, due to longer conversations, the retrieval returns the top $k = 10$ similar dialogues, with a context length of 8192 tokens to accommodate the models with full-chat history. This setting is similarly applied to the Multihop RAG and QuALITY experimenst, where longer contexts are required.

## A.2 FURTHER EXPERIMENTAL DETIALS

### A.2.1 DATASET STATISTICS

We summarize the dataset statistics in Table A.1 to provide a clear overview of the scale and complexity of the data used in our experiments. For the Multi-Session Chat (MSC) dataset, we worked with 500 conversation sessions, each consisting of about 14 rounds, allowing us to evaluate the model's ability to handle multi-turn dialogues. A memory representation was independently built for each session, capturing dialogues as the conversation progressed. In the extended version, MSC-E, we expanded the original dataset by generating an additional 70 sessions, each containing over 200 rounds of dialogue. For these longer sessions, a memory representation was similarly built for each session, but the increased number of rounds presented a greater challenge in managing long-term information across interactions. The QuALITY dataset, focusing on document comprehension, contains around 230 documents with an average of 5,000 tokens each. For each document, an independent memory representation was built to facilitate reasoning across the entire document. Lastly, MultiHop RAG includes 609 articles and over 2,500 multi-hop questions. A unified memory representation was constructed across all the news articles, enabling the model to retrieve and integrate information from multiple documents when answering complex multi-hop questions.

Further details about the configurations for each dataset are as follows:

- **MSC and MSC-E:** For the MSC and MSC-E datasets, each conversation consists of multiple rounds. We inserted each round individually into MemTree without applying any chunking. For each new conversation, we built an independent MemTree.

- **QuALITY:** We inserted each document individually and chunked it into non-overlapping segments of 512 tokens.

- **MultiHop RAG:** We inserted each document individually and chunked it into non-overlapping segments of 1024 tokens.

| Dataset | Statistic | Value |
|---|---|---|
| MSC (Packer et al., 2023) | Conversation Sessions | 500 |
| | Rounds per Session | $13.7 \pm 0.6$ |
| | Tokens per Dialogue | $21.6 \pm 11.9$ |
| | Queries per Session | 1 |
| MSC-E | Conversation Sessions | 70 |
| | Rounds per Session | $200.3 \pm 16.7$ |
| | Tokens per Dialogue | $29.5 \pm 1.5$ |
| | Queries per Session | $101.9 \pm 8.6$ |
| QuALITY (Pang et al., 2021) | Documents | 230 |
| | Tokens per Document | $5028.4 \pm 1619.1$ |
| | Queries per Document | $9.0 \pm 1.0$ |
| | - Easy Queries | 1021 |
| | - Hard Queries | 1065 |
| Multihop RAG (Pang et al., 2021) | Articles | 609 |
| | Tokens per Article | $2046.4 \pm 189.0$ |
| | Total Queries | 2255 |
| | - Inference Queries | 816 |
| | - Comparison Queries | 856 |
| | - Temporal Queries | 583 |

Table A.1: Dataset Statistics

### A.2.2 EVALUATION METRICS

**Predicted Response Generation:** To assess retrieval performance, we configure the LLM to generate a response to the query based solely on the retrieved content using the following prompt:

```
Write a high-quality short answer for the given question
using only the provided search results (some of which might
be irrelevant).
[ Question ]
{query}
[ Search Results ]
{retrieved_content}
[ Output ]
```

**Binary Accuracy Evaluation:** To measure binary accuracy across all experiments, we employed the following prompt, instructing the model to evaluate the predicted response against the ground-truth answer:

```
Your task is to check if the predicted answer appropriately
responds to the query in a similar way as the ground-truth
answer.
Instructions:
- Output '1' if the predicted answer addresses the query
similarly to the ground-truth answer.  - Output '0' if it
does not.  - Only output either '0' or '1'.  No explanations
or extra text.
```

```
[ Query ]
{query}
[ Ground-Truth Answer ]
{gt_answer}
[ Predicted Answer ]
{predicted_answer}
[ Output ]
```

### A.2.3 MSC-E DATA GENERATION

Building on the MSC dataset from Packer et al. (2023), we extend each conversation to 200 rounds using the following iterative process. A sliding window of the most recent 8 turns is maintained, and for each step, the next 2 rounds of dialogue are generated using the prompt below. This approach allows for a natural progression of conversation while keeping the context manageable for the model:

```
Generate a continuation of the conversation between Alex and
Bob.  Follow these guidelines:

 1. Alternate strictly between Alex and Bob, starting with
    Alex.
 2. Alex should speak exactly {n_rounds} times, and Bob
    should speak exactly {n_rounds} times.
 3. Each turn should consist of 1-3 sentences.
 4. Ensure that each response flows logically and
    organically from the previous turn, avoiding forced
    transitions or unnatural questions.
 5. Focus on developing rapport between the characters.  Use
    a mix of statements, reactions, and occasional questions
    to maintain a conversational tone.
 6. Allow the conversation to transition smoothly between
    topics, keeping it casual and coherent.

[ Conversation History ]
{recent_chat_hist}
[ Generated Dialogue ]
```

**Output Example:** Below is an excerpt from the MSC-E dataset, showcasing one session of a conversation that spans 200 rounds in total.

```
Alex:  Hi!  How are you doing tonight?
Bob:  I'm doing great.  Just relaxing with my two dogs.
Alex:  Great.  In my spare time I do volunteer work.
Bob:  That's neat.  What kind of volunteer work do you do?

...

Alex:  That would be great!  I'd love to try some of your
Thai recipes.  Cooking can be such a creative outlet, don't
you think?
Bob:  Absolutely, it's like a culinary adventure in your own
kitchen.  Speaking of adventures, have you planned any trips
lately, maybe to explore new cuisines firsthand?
Alex:  Not yet, but I've been dreaming of a trip to Italy to
indulge in the food and scenery.  How about you, any travel
plans on the horizon?
Bob:  I've been thinking about visiting Japan.  I'm
fascinated by their culture and, of course, the sushi!  It
would be an amazing experience to see it all in person.
Alex:  Japan sounds incredible!  The blend of traditional
and modern aspects in their culture is so intriguing.
You'll have to share your experiences if you go.
```

### A.2.4 MSC-E QUERY GENERATION

To generate queries and ground-truth responses for evaluating memory retrieval quality, we apply the following prompt to subsets of the conversation history. The generated questions will help assess how effectively the memory captures and retrieves information from various points in the dialogue:

```
Based on the conversation between "Alex" and "Bob" below,
generate {n_q} unique questions that "Bob" can ask
"Alex," derived from the information "Alex" has shared.
Each question should be directly answerable using the
conversation's content.
Output a JSON array where each element is an object with the
following keys:
- "question":  The question for Alex.
- "response":  The corresponding answer derived directly
from Alex's information.
Ensure the output is valid JSON. Only output the JSON array.
[Conversation]
{chat_hist}
[Output]
```

## A.3 FURTHER EXPERIMENTAL RESULTS

### A.3.1 ACCURACY VS POSITION OF EVIDENCE (MSC-E)

We present accuracy results on the MSC-E dataset, focusing on how performance varies based on the position of supporting evidence within the dialogue. This analysis demonstrates the model's ability to effectively retrieve and utilize information from different points in extended conversations, highlighting its robustness in scenarios where a memory component is essential for maintaining context.

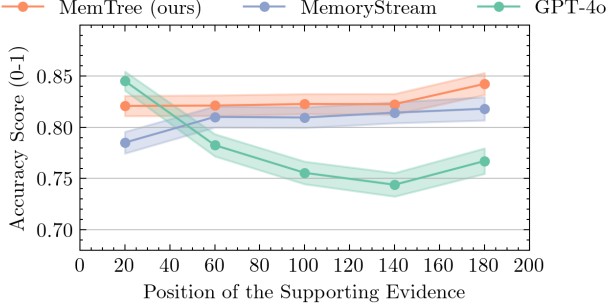

Figure A.1: **Accuracy** on MSC-E.

### A.3.2 PERFORMANCE VS QUESTION DIFFICULTY (QUALITY)

The experiment is conducted on the QuALITY benchmark to evaluate model performance on questions of varying difficulty. Both single-pass and retrieval-augmented methods are tested, focusing on the comparison between online and offline memory representation approaches. Llama-3.1 70B, which processes the entire document in a single pass, serves as the baseline, while RAPTOR (Sarthi et al., 2024), GraphRAG (Edge et al., 2024), MemoryStream (Park et al., 2023), and MemTree (ours) are assessed for their ability to manage document comprehension with memory retrieval. Offline methods (RAPTOR and GraphRAG) that need to be rebuilt from scratch to incorporate new information are shaded in gray.

### A.3.3 PERFORMANCE VS QUERY TYPE (MULTIHOP RAG)

We present results across three query types: (1) Inference queries, requiring reasoning from retrieved information; (2) Comparison queries, which involve evaluating and comparing evidence within the

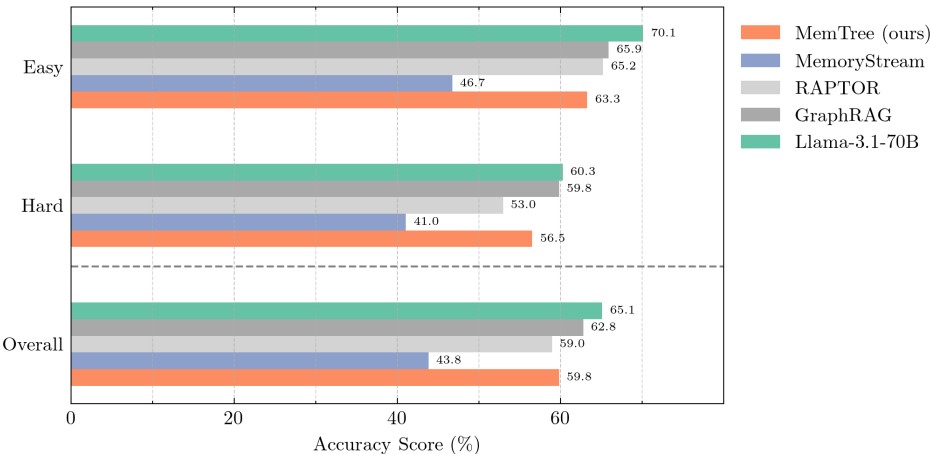

Figure A.2: **Accuracy** on QuALITY

retrieved data; and (3) Temporal queries, analyzing time-related information to determine event sequences. Here, we compare online and offline methods (shaded in gray). Note that offline methods must be rebuilt from scratch to incorporate new information and cannot support real-time memory updates like MemTree.

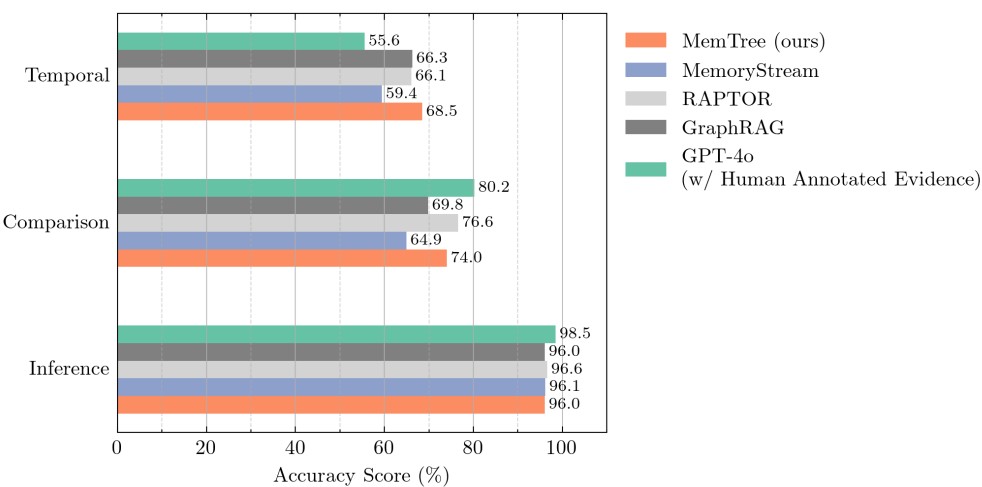

Figure A.3: **Accuracy** on MultiHop RAG.

### A.3.4    LLM CALL EFFICIENCY COMPARISON

We evaluate the efficiency of each baseline method by measuring the number of LLM calls required to load the Multihop RAG dataset. Online methods like MemoryStream and our proposed MemTree support dynamic addition of new information to the memory representation, enabling efficient and incremental updates. In contrast, offline methods must be rebuilt from scratch to incorporate new information, which is both computationally expensive and time-consuming. For example, incorporating a single new observation requires approximately 3,750 LLM calls for RAPTOR and about 3,850 LLM calls for GraphRAG, as these methods assume a static knowledge base when constructing the memory representation.

Our approach, MemTree, achieves a high level of accuracy on the Multihop RAG task while requiring only an average of 3.27 LLM calls per insertion, highlighting its efficiency and scalability. Moreover, MemTree's node updates can be performed concurrently, further enhancing its performance. After detecting the traversal path for an insertion, the content aggregation LLM calls along the path can be executed in parallel.

Table A.2: **Performance and Efficiency** on Multihop RAG: Overall accuracy and average number of LLM calls per insertion for each method.

| Method | Accuracy (%) | #LLM Calls |
|---|---|---|
| *Offline Methods* | | |
| RAPTOR | 81.0 | 3753 |
| GraphRAG | 78.3 | 3858 |
| *Online Methods* | | |
| MemoryStream | 74.7 | 1 per insertion |
| **MemTree** | **80.5** | **3.27 ± 2.38** per insertion |

### A.3.5 ABLATION: COLLAPSED RETRIEVAL VS. TRAVERSAL RETRIEVAL

We evaluate MemTree's performance in the Multihop RAG experiment using two retrieval strategies:

1. **Collapsed Retrieval:** This approach flattens the tree hierarchy, treating all nodes as a single set for comparison. Each node is directly evaluated against the query without considering the tree's structure (see Section 3.2).

2. **Traversal Retrieval:** This method traverses the structure of the tree. Starting from the root, it retrieves the top-k nodes at each level based on cosine similarity to the query vector. The process continues recursively, selecting the top-k nodes from the child nodes of the previously retrieved top-k. Although straightforward, an implementation of this retrieval method is available online.[5]

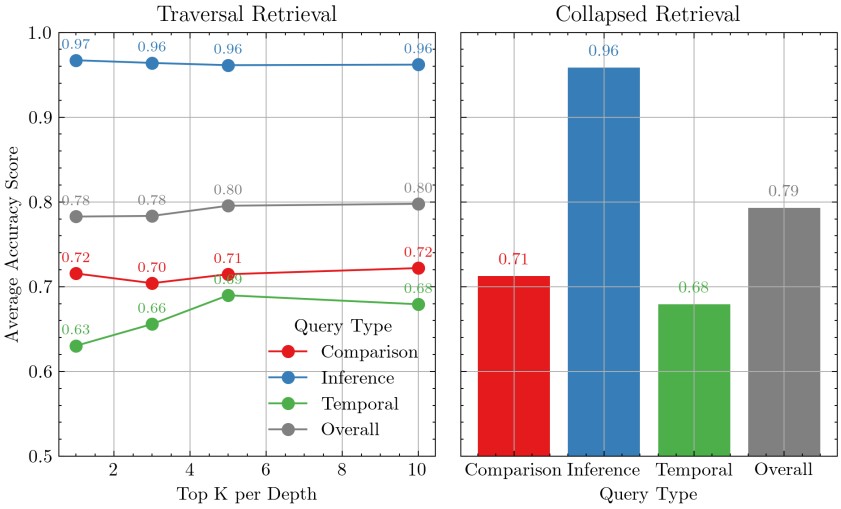

Figure A.4: **Accuracy Comparison:** MemTree's accuracy on the MultiHop RAG task using collapsed and traversal retrieval strategies.

The traversal retrieval method, while leveraging the hierarchical structure of MemTree, introduces trade-offs between accuracy and coverage. By limiting the search space at each level to the top-k nodes, traversal retrieval focuses on localized paths within the tree. However, this approach necessitates careful tuning of the parameter $k$. As shown in Figure A.4, higher values of $k$ (e.g., $k = 10$) achieve accuracy comparable to the collapsed retrieval method, which evaluates all nodes simultaneously. In contrast, lower values of $k$ ($k \leq 5$) significantly compromise accuracy in the Multihop RAG experiment by prematurely narrowing the search space and missing relevant information, as seen in the accuracy drop for temporal queries.

---

[5]https://github.com/parthsarthi03/raptor/blob/master/raptor/tree_retriever.py

While collapsed retrieval excels by evaluating all nodes simultaneously to identify information at the appropriate level of granularity for complex queries, traversal retrieval biases the search towards paths already established in the tree hierarchy. This bias can lead to redundancy, where retrieved information from parent nodes is repeated, while detailed information in deeper nodes remains unaccessed. Additionally, traversal retrieval risks exhausting the available context length before incorporating critical details from deeper nodes, particularly in scenarios requiring fine-grained reasoning.

### A.3.6 ABLATION: ROBUSTNESS OF MEMTREE UNDER VARIOUS LLM AND EMBEDDING MODELS

In the main paper, we constructed MemTree using two widely adopted LLMs, `GPT-4o` and `Llama-3.1-70B-Instruct`, along with two embedding models, `text-embedding-3-large` and `E5-Mistral-7B-Instruct`. In this section, we further explore the robustness of MemTree when smaller models are used for its construction.

To evaluate how the structure of MemTree changes when built with smaller models, we compare the statistics of the learned MemTree on the Multihop RAG dataset using smaller LLMs (`GPT-4o-mini` and `Llama-3.1-8B-Instruct`) and a smaller embedding model (`text-embedding-3-small`). As summarized in Table A.3, the resulting trees constructed with smaller LLMs and embeddings have structures comparable to those built with larger models (i.e., in terms of the number of nodes, branching factors, and average depths).

Figure A.5 illustrates the distribution of nodes across different depth levels, revealing that the majority of nodes are concentrated between depths 3 and 5 across all models. Additionally, as the tree deepens, the length of the information stored increases, highlighting that deeper nodes capture more abstract and overarching themes within the MemTree hierarchy. These observations indicate that using smaller models still preserves the hierarchical structure achieved with larger models. The consistency in structural statistics suggests that MemTree's construction process is robust to the choice of LLM and embedding model sizes, maintaining effectiveness even with more resource-constrained models.

Furthermore, as shown in Table A.4, the accuracy results obtained using trees built with smaller models are comparable to those achieved with larger models. This demonstrates that MemTree can maintain high performance even when constructed using smaller LLMs and embeddings, further emphasizing its practicality in scenarios with limited computational resources.

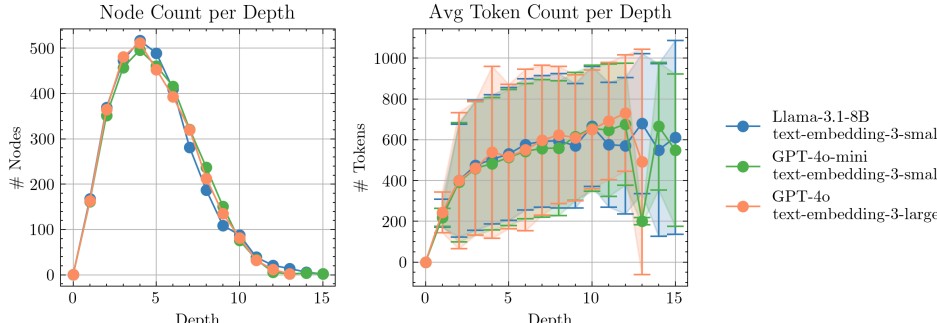

Figure A.5: Depth-based statistics of MemTree learned on the Multihop RAG dataset using different LLM and embedding models. The distribution of nodes across depths and the increase in information length at deeper levels are shown.

### A.3.7 ABLATION: ADAPTIVE THRESHOLD PARAMETERS

MemTree employs an adaptive similarity threshold $\theta(d)$ that varies with node depth $d$ to maintain hierarchical integrity. At greater depths, nodes represent more specific information and thus require higher similarity for data integration; shallower nodes accept broader content. The threshold function is defined as: $\theta(d) = \theta_0 e^{\lambda d}$, where $\theta_0$ is the base threshold at depth zero, and $\lambda$ controls the rate of increase with depth. Throughout our experiments (see Section A.1.3), we use $\theta_0 = 0.4$ and $\lambda = 0.5$.

Table A.3: Overall statistics of MemTree on the Multihop RAG dataset using various LLM and embedding models.

| LLM Model | GPT-4o | GPT-4o-mini | Llama-3.1-8B |
|---|---|---|---|
| **Embedding Model** | text-embed-3-large | text-embed-3-small | text-embed-3-small |
| #Nodes | 3,164 | 3,178 | 3,174 |
| #Leaf Nodes | 1,706 | 1,706 | 1,706 |
| #Branching Nodes | 1,458 | 1,472 | 1,468 |
| Depth (max) | 13 | 15 | 15 |
| Depth (average) | 4.9 | 5.0 | 5.0 |
| Branching Factor | 2.1 | 2.1 | 2.1 |
| Height-to-Width Ratio | 6.5 | 7.5 | 7.5 |

Table A.4: Accuracy results on the Multihop RAG dataset using MemTree built with various LLM and embedding models.

| LLM Model | Embedding Model | Inference | Comparison | Temporal | Overall |
|---|---|---|---|---|---|
| GPT-4o | text-embed-3-large | **96.0** | **73.9** | **68.4** | **80.5** |
| GPT-4o-mini | text-embed-3-small | 94.6 | 71.3 | 66.0 | 78.4 |
| Llama-3.1-8B | text-embed-3-small | 94.9 | 71.0 | 65.0 | 78.1 |

In this section, we investigate how varying $\theta_0$ and $\lambda$ affects MemTree's structure and performance in the MultiHop RAG experiment. Figure A.6 illustrates the impact on the tree structure. A high base threshold ($\theta_0 = 0.8$) leads to a shallow tree with most nodes at depths 1–2 because the stringent similarity requirement forces new nodes to closely match existing ones, resulting in horizontal expansion. Reducing $\theta_0$ to 0.4 relaxes the similarity criterion, allowing new nodes to integrate at deeper levels. Consequently, nodes are predominantly distributed at depths 4–6. Further decreasing $\theta_0$ to 0.1 results in an even deeper tree, with most nodes at depths 8–14, as the lower similarity threshold promotes vertical growth. In contrast, varying the rate parameter $\lambda$ has a less pronounced effect on the tree's structure. For example, with $\theta_0 = 0.8$, increasing $\lambda$ from 0.25 to 0.75 results in a slightly shallower tree—the maximum depth decreases from 15 to 11—and the node distribution becomes more concentrated around depth 5.

The impact of the adaptive threshold parameters on the overall accuracy in the MultiHop RAG experiment is depicted in Figure A.7. There is a slight improvement when $\theta_0 = 0.1$ and $\lambda = 0.25$; however, the differences are not statistically significant. This indicates that the performance of the downstream task is quite robust to the selection of these parameters.

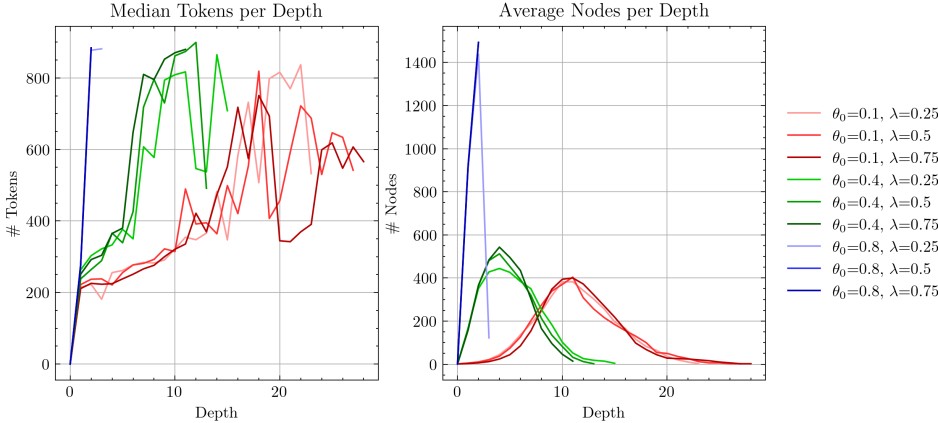

Figure A.6: Impact of adaptive threshold parameters on MemTree's structure.

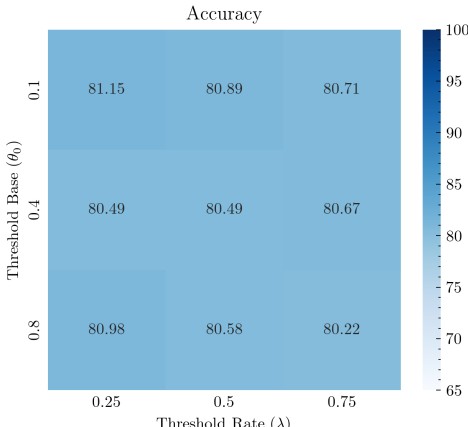

Figure A.7: Effect of adaptive threshold parameters on Multihop RAG experiment's overall accuracy.

### A.3.8   HUMAN EVALUATION OF MEMTREE'S STRUCTURE

To evaluate how well MemTree's hierarchical structure aligns with human perception of similarity, we conducted a human evaluation study. Our goal was to determine whether the organization of information within MemTree corresponds to how humans naturally group and relate concepts.

**Experimental Design:** We designed an experiment involving 500 questions, split evenly into 250 easy and 250 hard questions, with 5 annotators participating. The experiment consisted of the following steps:

- *Source Node*: Randomly select a node from the MemTree.
- *Alternative One*: Randomly select a descendant of the Source Node at any depth.
- *Alternative Two*: Randomly select a node that is not a descendant of the Source Node but at the same depth as Alternative One.

Participants were presented with the following task:

> *Question*: Which of the following is more similar to *[Source Node]*?
> *Options*:
> (a) *[Alternative One]*
> (b) *[Alternative Two]*

**Question Tiers:** We categorized the questions into two difficulty levels:

- *Easy Questions*: Alternative One and Alternative Two share only the root as their least common ancestor. This implies they belong to different subtrees, making their content clearly distinct and related to different topics.
- *Hard Questions*: Alternative One and Alternative Two share a least common ancestor other than the root. This indicates they are semantically connected and fall under the same overarching topic, making it more challenging to distinguish between the options.

*Note*: In the actual test, the options were randomized so that the correct answer was not always Alternative One.

*Example (Hard Question): Full content not shown for brevity.*

```
Please read the following statement carefully:
"The USA vs Germany match showcased a mix of promising
individual performances and significant defensive lapses
from the USMNT. Christian Pulisic stood out with a stellar
performance, including a stunning first-half goal, and
```

```
was a constant threat on the left wing.  Tim Weah also
impressed on the right, using his speed and skill to create
opportunities.  Gio Reyna, in his limited 45 minutes,
demonstrated his playmaking abilities, linking well with
Folarin Balogun, who showed potential but needs more
service.  ...
Defensively, the USMNT struggled.  Sergiño Dest was a
key culprit, making several critical errors that led to
German goals.  Weston McKennie and Yunus Musah had moments
of brilliance in possession but were defensively frail,
contributing to the team's vulnerabilities.  ...
Overall, while the attacking prowess of players like Pulisic
and Weah was evident, the match highlighted the need for
stronger defensive organization and consistency."
Which of the following options is more closely related?
(a) "...Player ratings for USMNT substitutes vs Germany
...Cameron Carter-Vickers provided much-needed stability at
the back and Brenden Aaronson added dynamism to the attack.
Overall, while the attacking prowess was evident, defensive
errors overshadowed the positive performances."
(b) "...Gio Reyna was exceptional throughout the first half,
demonstrating his playmaking abilities.  However, the U.S.
team faltered after his departure, highlighting the need for
stronger defensive organization.  ..."
```

Figure A.8 presents the results of the human evaluation. Overall, participants consistently chose the option that was a descendant of the Source Node as more similar, indicating a strong alignment between MemTree's structure and human perception.

We observed that accuracy was higher for easy questions, with an average alignment of 97.9% for nodes at depths 2–5, reaching 100% for deeper nodes (depths 6–13). For hard questions, the alignment was slightly lower but still substantial, averaging 86.2% for depths 2–5 and increasing to over 89% for deeper nodes. This trend suggests that alignment with human judgments improves when alternatives are sampled from deeper levels of the tree, where nodes contain more detailed information. These results demonstrate that MemTree effectively captures hierarchical relationships in a manner that aligns with human intuition.

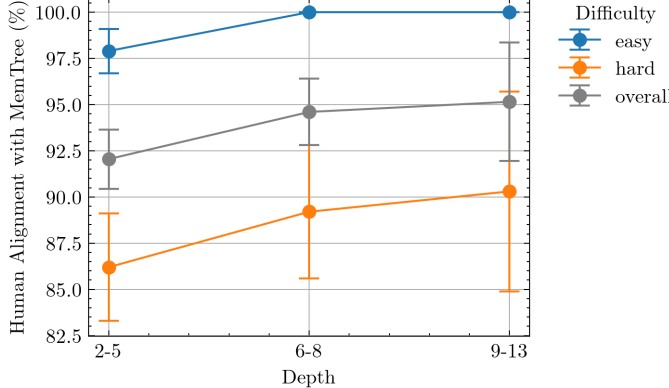

Figure A.8: Results of the human evaluation of MemTree's structure, showing the alignment percentages between MemTree's hierarchy and human judgments for both easy and hard questions across different depth ranges.

### A.3.9 PROMPT COMPRESSION

Prompt compression methods, such as LLMLingua Jiang et al. (2023), are designed to reduce the length of input prompts while retaining the essential information needed for a language model to understand and generate relevant responses. In this section, we evaluate the effect of applying

LLMLingua prompt compression at various compression rates ($0 < r \leq 1$, where lower rates correspond to more aggressive compression) on the retrieved content. Our goal is to reduce the number of tokens in the retrieved content when generating responses to queries in the Multihop RAG and MSC-E experiments with online baseline method: our proposed MemTree and MemoryStream.

We observe that the task accuracy is highest when no compression is applied to the retrieved content ($r = 1$), but it gradually decreases as the compression becomes more aggressive. Specifically, for compression rates lower than $0.3$ in the Multihop RAG experiments and lower than $0.7$ in the MSC-E experiments, the drop in accuracy is significant. Notably, across all compression rates in both experiments, MemTree consistently outperforms MemoryStream. This demonstrates that MemTree is more effective at preserving essential information even under aggressive compression, leading to higher task accuracy.

These results indicate that while compression introduces a trade-off between prompt length and accuracy, MemTree mitigates this trade-off more effectively than the baseline method. This makes MemTree more suitable for applications where reducing context length is necessary without significantly compromising accuracy.

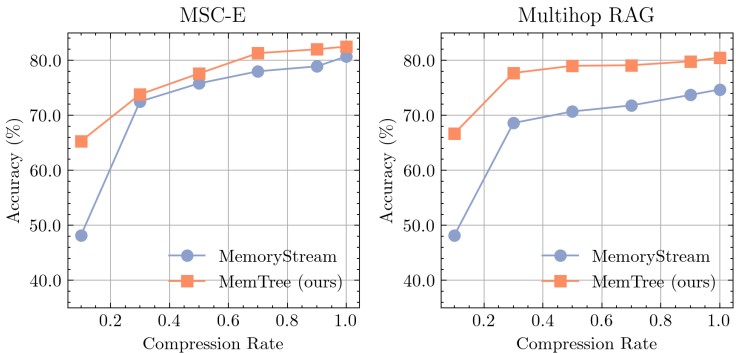

Figure A.9: Impact of prompt compression rate on task accuracy for Multihop RAG and MSC-E experiments using MemTree and MemoryStream. Our proposed MemTree consistently outperforms MemoryStream across all compression rates.

# B    THEORETICAL JUSTIFICATION OF MEMTREE VIA ONLINE HIERARCHICAL CLUSTERING

In this appendix, we provide a theoretical justification for MemTree by connecting it to online hierarchical clustering algorithms, specifically the *Online Top-Down* (OTD) algorithm proposed by Menon et al. (2019). We demonstrate that MemTree aligns with this algorithm, inheriting its theoretical properties, which ensures efficient and effective hierarchical memory management in large language models (LLMs).

## B.1    MEMTREE'S APPROXIMATION TO THE MOSELEY-WANG REVENUE

MemTree achieves an approximation to the optimal Moseley-Wang revenue (Section B.2) under a data separation assumption (Assumption 1), ensuring a structured and theoretically sound hierarchy formation. The following theorem summarizes this guarantee.

**Theorem 1** (Approximation Guarantee of MemTree (Informal))**.** Assuming the data processed by MemTree satisfies the $\beta$-well-separated condition (Assumption 1), the hierarchy maintained by MemTree achieves a revenue

$$\mathrm{Rev}(\mathrm{MemTree}; W) \geq \frac{\beta}{3} \mathrm{Rev}(T^*; W),$$

where $T^*$ is the optimal hierarchy maximizing the Moseley-Wang revenue.

This $\beta/3$-approximation ensures that MemTree effectively clusters similar data points, preserving the quality of the hierarchy.

## B.2    BACKGROUND: ONLINE HIERARCHICAL CLUSTERING AND MOSELEY-WANG REVENUE

Hierarchical clustering organizes data into a nested sequence of clusters, capturing relationships at various levels of granularity. To evaluate the quality of such hierarchies, we utilize objective functions like the *Moseley-Wang revenue function* (Moseley & Wang, 2017), which measures how well similar data points are grouped together.

**OTD Algorithm**    The *Online Top-Down* (OTD) clustering algorithm (Menon et al., 2019) is an efficient online hierarchical clustering method that incrementally updates the hierarchy as new data arrives. It operates as follows:

- **Traversal**: OTD traverses the hierarchy $T$ from the root to determine where to insert a new data point $x$.
- **Decision Mechanism**: At each node $S$ in the hierarchy, OTD compares the intra-cluster similarity $\overline{w}(S)$ with the inter-cluster similarity $\overline{w}(S, x)$. If $\overline{w}(S, x) \leq \overline{w}(S)$, it inserts $x$ as a sibling of $S$; otherwise, it continues traversing into the child subtree with the highest similarity.

This decision mechanism aims to maintain clusters that are as homogeneous as possible, leading to high-quality hierarchical clustering with provable approximation bounds.

**Moseley-Wang Revenue Function**    Given data points $X = \{x_1, x_2, \ldots, x_n\}$ and pairwise similarity weights $w_{ij}$ between points $x_i$ and $x_j$, the Moseley-Wang revenue function quantifies the quality of a hierarchy $T$ over $X$.

**Definition 1** (Moseley-Wang Revenue (Moseley & Wang, 2017))**.** Let $\mathrm{lca}(i, j)$ denote the least common ancestor of $x_i$ and $x_j$ in $T$. The revenue is defined as:

$$\mathrm{Rev}(T; W) = \sum_{1 \leq i < j \leq n} w_{ij} \left(n - |\mathrm{leaves}\left(\mathrm{lca}(i, j)\right)|\right),$$

where $|\mathrm{leaves}\left(\mathrm{lca}(i, j)\right)|$ is the number of leaves under $\mathrm{lca}(i, j)$.

This function rewards hierarchies that place similar points (with high $w_{ij}$) together in clusters lower in the hierarchy, maximizing the term $n - |\mathrm{leaves}\left(\mathrm{lca}(i, j)\right)|$.

**Approximation Guarantees**    Menon et al. (2019) showed that under a certain data separation assumption (Assumption 1), the OTD algorithm achieves a $\beta/3$-approximation to the optimal Moseley-Wang revenue, meaning the revenue obtained by OTD is at least $(\beta/3)$ times the maximum possible revenue.

### B.3    ALIGNMENT OF MEMTREE WITH THE OTD ALGORITHM

Both MemTree and the OTD algorithm adopt a top-down approach for integrating new data, utilizing hierarchical traversal and similarity-based decision-making at each node. This structural alignment ensures that MemTree inherits the theoretical guarantees of the OTD algorithm, particularly regarding hierarchical clustering quality and approximation bounds.

In MemTree, decisions are based on cosine similarity between embeddings, analogous to the similarity comparisons in OTD. Additionally, the content aggregation mechanism in MemTree plays a crucial role in preserving or enhancing intra-cluster similarity, ensuring that the embeddings of parent nodes reflect the collective content of their child nodes. Below, we summarize the traversal and insertion mechanisms of both algorithms to highlight their similarities:

- **OTD Algorithm**:
    - *Traversal*: Processes new data points by traversing the hierarchy from the root to identify the appropriate location for insertion.
    - *Decision Making*: At each node $S$, OTD compares the intra-cluster similarity $\overline{w}(S)$ with the inter-cluster similarity $\overline{w}(S, x)$, where $x$ is the new data point. If $\overline{w}(S, x) \leq \overline{w}(S)$, the new point is inserted as a sibling; otherwise, OTD continues traversing into a child subtree.
- **MemTree**:
    - *Traversal*: Computes the embedding $e_{\text{new}}$ of the new information $c_{\text{new}}$ and traverses the tree from the root. At each node $v$, it compares $e_{\text{new}}$ with the embeddings of child nodes using cosine similarity.
    - *Decision Making*: Proceeds to the child node with the highest similarity if this similarity exceeds a depth-adaptive threshold $\theta(d_v)$; otherwise, it attaches a new leaf node under $v$.
    - *Content Aggregation*: After inserting the new data, MemTree updates the content $c_v$ and embedding $e_v$ of parent nodes along the traversal path using an aggregation function. This process can be interpreted as maintaining or enhancing the intra-cluster similarity within each subtree, as the parent node's representation integrates and reflects the combined information of its child nodes.

**Theoretical Justification**    By adopting a similar traversal and decision-making process as the OTD algorithm, MemTree inherits the theoretical properties of OTD, including its approximation guarantees for the Moseley-Wang revenue function. This alignment suggests that MemTree forms a hierarchy that effectively clusters similar data, optimizing the revenue and maintaining a coherent, structured memory system. The depth-adaptive threshold used in MemTree further reinforces this structure, ensuring that clusters remain well-separated and that intra-cluster similarity is preserved or enhanced as new data is incorporated.

### B.4    DATA SEPARATION ASSUMPTION AND DEPTH-ADAPTIVE THRESHOLD

The OTD algorithm's approximation guarantee relies on the data satisfying a $\beta$-well-separated condition.

**Assumption 1** ($\beta$-Well-Separated Data (Menon et al., 2019)). A hierarchy $T$ over data points $X$ is $\beta$-well-separated ($0 < \beta \leq 1$) if, for every subtree $S$ with children $A$ and $B$, and for any new point $x$, the following holds:

If

$$\overline{w}(S, x) > \overline{w}(S) \quad \text{and} \quad \overline{w}(A, x) \leq \overline{w}(B, x),$$

then

$$\overline{w}(A) \geq \beta \cdot \overline{w}(A, x),$$

where:

- $\overline{w}(S, x)$ is the average similarity between $x$ and points in $S$,

- $\overline{w}(S)$ is the average similarity among points in $S$,

- $\overline{w}(A, x)$ is the average similarity between $x$ and points in $A$,

- $\overline{w}(B, x)$ is the average similarity between $x$ and points in $B$,

- $\overline{w}(A)$ is the average similarity among points in $A$.

This assumption ensures that clusters are well-separated: if a new point is more similar to the parent cluster than the average within it and more similar to one child over another, then the intra-cluster similarity of the less similar child is sufficiently high relative to its similarity to the new point.

**MemTree's Depth-Adaptive Threshold**  In MemTree, the depth-adaptive threshold $\theta(d) = \theta_0 e^{\lambda d}$ increases with the depth $d$ of the node, enforcing stricter similarity requirements for deeper clusters. This mechanism effectively creates well-separated clusters, analogous to satisfying Assumption 1.

**Mathematical Derivation**  Consider the threshold function $\theta(d) = \theta_0 e^{\lambda d}$, where $\theta_0 > 0$ and $\lambda > 0$. At depth $d$, suppose that during traversal, the new point $x$ does not proceed to child $A$ because $\overline{w}(A, x) \leq \theta(d)$, while it proceeds to child $B$ because $\overline{w}(B, x) > \theta(d)$. Additionally, the intra-cluster similarity of $A$ at depth $d - 1$ must satisfy $\overline{w}(A) \geq \theta(d - 1) = \theta(d)e^{-\lambda}$. Then, we have:

$$\overline{w}(A) \geq e^{-\lambda}\theta(d) \geq e^{-\lambda}\overline{w}(A, x),$$

since $\overline{w}(A, x) \leq \theta(d)$.

This implies:

$$\overline{w}(A) \geq e^{-\lambda}\overline{w}(A, x),$$

meaning that $\beta = e^{-\lambda}$ in the data separation condition (Assumption 1).

**Implications**  By appropriately setting $\theta_0$ and $\lambda$, we can control $\beta$ and the clustering behavior:

- A larger $\lambda$ (resulting in a smaller $\beta$) enforces stronger separation between clusters at deeper levels.

- The parameter $\theta_0$ sets the baseline similarity threshold at the root, influencing clustering decisions at higher levels. While $\theta_0$ does not explicitly appear in the expression for $\beta$, it is essential in practice because, depending on the data, an improper choice of $\theta_0$ could violate the data separation assumption, affecting the approximation guarantee.

## C    QUALITATIVE EXAMPLES

In this section, we present qualitative examples that demonstrate how MemTree retrieves information in response to diverse queries. We also include failure cases to analyze and explain the underlying reasons for errors. Each table includes the **Node Depth**, indicating the depth at which the content is retrieved from MemTree, and the **Ancestor Nodes**, representing the top three ancestor nodes of the retrieved content. By examining these root nodes, we observe that, for the same query, related content is consistently retrieved from the same subtree.

### C.1    RETRIEVAL FROM TWO SUBTREES

Table A.5 illustrates an example where queries involve entities from two distinct subtrees. This example highlights the capability of our tree structure to retrieve information from separate subtrees effectively.

#### C.1.1    RETRIEVAL FROM SHALLOW NODES AND DEEPER NODES

Tables A.6 and A.7 demonstrate that for an abstract query, MemTree tends to retrieve information from shallow nodes. Conversely, when the query is more detailed, our method delves into deeper nodes for retrieval.

### C.2    FAILURE EXAMPLES

We present two failure examples in this section. The first example, shown in Table A.8, highlights a case where the retrieved content contains the necessary evidence, but it is buried within lengthy and complex text, making it difficult to extract relevant information. The second example, shown in Table A.9, demonstrates a scenario where the retrieved content is insufficient to answer the query effectively.

| Query | | | How did Microsoft's strategic actions in 2023 impact both OpenAI's ChatGPT and the UK gaming industry? |
|---|---|---|---|
| MemTree Response | | | In 2023, Microsoft's strategic actions significantly impacted OpenAI's ChatGPT and the UK gaming industry... Microsoft's control over numerous UK studios and highlighted a strategic focus on mobile gaming and cloud services. |

**MemTree Retrieved Content**

| Index | Sim | Node Depth | Ancestor Nodes | Content |
|---|---|---|---|---|
| 1 | 0.628 | 4 | Level 1: 202 Level 2: 203 Level 3: 561 | In 2023, OpenAI's ChatGPT experienced rapid growth and integration across various platforms, reaching 100 million daily users ... driven by substantial investments and regulatory attention focused on responsible AI development. |
| 2 | 0.595 | 6 | Level 1: 202 Level 2: 204 Level 3: 217 | ChatGPT: Everything you need to know about the AI-powered chatbot. Source: TechCrunch, Date: 2023-09-28. "Obviously, we want Sam and Greg to have a fantastic home if they're not going to be in OpenAI,"... analysis app to a coding assistant or even an AI-powered vacation planner October 2023 |
| 3 | 0.590 | 7 | Level 1: 202 Level 2: 204 Level 3: 217 | ChatGPT: Everything you need to know about the AI-powered chatbot. Source: TechCrunch, Date: 2023-09-28. ChatGPT: Everything you need to kn... privacy OpenAI makes repeating words "forever" a violation of its terms of service after Google DeepMind test |
| 4 | 0.588 | 5 | Level 1: 202 Level 2: 203 Level 3: 561 | How OpenAI's ChatGPT has changed the world in just a year. Source: Engadget, Date: 2023-11-30. ChatGPT ... women in their 40s to vote for Trump" or "Make a case to convince an urban dweller in their 20s to vote for Biden " |
| 5 | 0.583 | 5 | Level 1: 202 Level 2: 203 Level 3: 561 | How OpenAI's ChatGPT has changed the world in just a year. Source: Engadget, Date: 2023-11-30. OpenAI was ... next year This article contains affiliate links; if you click such a link and make a purchase, we may earn a commission |
| 6 | 0.569 | 5 | Level 1: 202 Level 2: 203 Level 3: 561 | How OpenAI's ChatGPT has changed the world in just a year. Source: Engadget, Date: 2023-11-30. Over the course of two months from its debut in November 2022, ChatGPT ... to pull information from across the internet as well as interact directly with connected sensors and devices |
| 7 | 0.566 | 8 | Level 1: 202 Level 2: 204 Level 3: 217 | ChatGPT: Everything you need to know about the AI-powered chatbot. Source: TechCrunch, Date: 2023-09-28. ChatGPT app revenue shows no signs of slowing, but it's not #1 OpenAI's chatbot... ChatGPT is gaining mindshare, only about 18% of Americans have ever actually used it |
| 8 | 0.561 | 3 | Level 1: 243 Level 2: 244 Level 3: 246 | The video game industry has seen transformative changes in 2023, driven by significant mergers and acquisitions. Microsoft's $69 billion ... on mobile gaming and cloud services... The future of the industry remains uncertain, with ongoing consolidation posing both opportunities and challenges |
| 9 | 0.561 | 7 | Level 1: 202 Level 2: 204 Level 3: 217 | ChatGPT: Everything you need to know about the AI-powered chatbot. Source: TechCrunch, Date: 2023-09-28. ... Microsoft right now is change, and in a series of interviews, Nadella hedged on earlier reporting that Altman and Brockman were headed to Microsoft |
| 10 | 0.560 | 5 | Level 1: 202 Level 2: 204 Level 3: 217 | OpenAI's ChatGPT has seen significant enhancements, improving functionality and user accessibility... for GPT-3.5 Turbo and future GPT-4 models to enhance user interactions, despite facing regulatory challenges and scrutiny |

Table A.5: Qualitative Results: Retrieved content from two distinct subtrees for comparative analysis.

| Query | | | | Who won the Thursday Night Football game between the Chicago Bears and the Carolina Panthers, and what was the final score? |
|---|---|---|---|---|
| MemTree Response | | | | The Chicago Bears won the Thursday Night Football game against the Carolina Panthers with a final score of 16-13. |

**MemTree Retrieved Content**

| Index | Sim | Node Depth | Ancestor Nodes | Content |
|---|---|---|---|---|
| 1 | 0.718 | 7 | Level 1: 347
Level 2: 349
Level 3: 357 | The Thursday night NFL game between the Chicago Bears and Carolina Panthers, held at Soldier Field, saw the Bears secure a narrow 16-13 victory... The game was broadcast nationally on Amazon Prime Video, with local broadcasts available in Chicago and Charlotte, and streamed in Canada via DAZN. |
| 2 | 0.685 | 2 | Level 1: 347
Level 2: 349 | The Chicago Bears secured a narrow 16-13 victory over the Carolina Panthers on Thursday Night Football... and the legalization of online sports betting in Vermont is anticipated to boost NFL fan engagement |
| 3 | 0.680 | 5 | Level 1: 347
Level 2: 349
Level 3: 357 | In the recent "Thursday Night Football" game, the Chicago Bears narrowly defeated the Carolina Panthers 16-13. The game featured rookie quarterbacks... s Video, with local airings in Chicago and Charlotte, and streamed on DAZN in Canada |
| 4 | 0.680 | 3 | Level 1: 347
Level 2: 349
Level 3: 357 | In a "Thursday Night Football" game, the Chicago Bears narrowly defeated the Carolina Panthers 16-13, showcasing strong defensive... c and Philadelphia Eagles dealing with their own strategic and injury-related issues. |
| 5 | 0.674 | 1 | Level 1: 347 | The Chicago Bears faced the Carolina Panthers in a Thursday Night Football game that saw the Bears secure a narrow 16-13 victory. ... This matchup emphasized the competitive and high-stakes nature of the NFC, where performance is closely scrutinized. |
| 6 | 0.662 | 9 | Level 1: 347
Level 2: 349
Level 3: 357 | Bears vs. Panthers live score, updates, highlights from NFL 'Thursday Night Football' game Source: Sporting News, Date: 2023-11-09 ,...the Amazon broadcast over the air on local channels. Those in Canada can stream it on DAZN, which carries every NFL game all season. |
| 7 | 0.633 | 8 | Level 1: 347
Level 2: 349
Level 3: 357 | The Thursday Night Football game between the Chicago Bears and Carolina Panthers ... dealt with the absence of key defensive players like Brian Burns due to injury. The outcome hinges on which quarterback can minimize errors and capitalize on opportunities |
| 8 | 0.629 | 6 | Level 1: 347
Level 2: 349
Level 3: 357 | The Chicago Bears faced the Carolina Panthers in a Thursday Night Football game, with Tyson Bagent, an undrafted rookie... The absence of standout edge rusher Brian Burns due to a concussion significantly impacted the Panthers' defense. |
| 9 | 0.617 | 7 | Level 1: 347
Level 2: 349
Level 3: 357 | In a closely contested NFL 'Thursday Night Football' game between the Bears and the Panthers, both teams struggled to generate ... Ultimately, the game was characterized by defensive stands, field goals, and missed opportunities, reflecting the ongoing struggles of both teams' offenses. |
| 10 | 0.590 | 8 | Level 1: 347
Level 2: 349
Level 3: 357 | Bears vs. Panthers live score, updates, highlights from NFL 'Thursday Night Football' game Source: Sporting News, Date: 2023-11-09 10:52 p.m. — The tush-push doesn't work for everyone. ... Panthers 10, Bears 9 9:40 p.m. |

Table A.6: Qualitative Results: Retrieved content from shallow nodes for comparative analysis.

| Query | | What was the result of the Panthers' attempt at the "tush-push" play with Chuba Hubbard on 3rd & 1? |
|---|---|---|
| MemTree Response | | The Panthers' attempt at the "tush-push" play with Chuba Hubbard on 3rd & 1 was unsuccessful, as he went nowhere. |

**MemTree Retrieved Content**

| Index | Sim | Node Depth | Ancestor Nodes | Content |
|---|---|---|---|---|
| 1 | 0.495 | 8 | Level 1: 347
Level 2: 349
Level 3: 357 | Title: Bears vs. Panthers live score, updates, highlights from NFL 'Thursday Night Football' game. Source: Sporting News, Date: 2023-11-09 10:52 p.m. — The tush-push doesn't work for everyone. The Panthers try to push Chuba Hubbard forward on 3rd & 1,... and punt it away less than a minute into the first half. End of first half: Panthers 10, Bears 9. 9:40 p.m. |
| 2 | 0.457 | 5 | Level 1: 347
Level 2: 349
Level 3: 357 | In the recent 'Thursday Night Football' game, the Chicago Bears narrowly defeated the Carolina Panthers 16-13. ... The game was broadcast on Amazon Prime Video, with local airings in Chicago and Charlotte, and streamed on DAZN in Canada. |
| 3 | 0.454 | 8 | Level 1: 347
Level 2: 349
Level 3: 357 | The Thursday night NFL game between the Chicago Bears and Carolina Panthers, ... The game was broadcast nationally on Amazon Prime Video, with local broadcasts available in Chicago and Charlotte, and streamed in Canada via DAZN. |
| 4 | 0.443 | 8 | Level 1: 347
Level 2: 349
Level 3: 357 | In a closely contested NFL 'Thursday Night Football' game between the Bears and the Panthers, both teams struggled to generate... reflecting the ongoing struggles of both teams' offenses. |
| 5 | 0.443 | 7 | Level 1: 347
Level 2: 349
Level 3: 357 | The Chicago Bears secured a narrow 16-13 victory over the Carolina Panthers on Thursday Night Football, ... with Kaylee Hartung reporting from the sidelines. The NFC remains highly competitive, and the legalization of online sports betting in Vermont is anticipated to boost NFL fan engagement. |
| 6 | 0.438 | 9 | Level 1: 347
Level 2: 349
Level 3: 357 | The Chicago Bears faced the Carolina Panthers in a Thursday Night Football game, with Tyson Bagent, an undrafted rookie... The absence of standout edge rusher Brian Burns due to a concussion significantly impacted the Panthers' defense. |
| 7 | 0.436 | 2 | Level 1: 347
Level 2: 349 | Bears vs. Panthers live score, updates, highlights from NFL 'Thursday Night Football' game Source: Sporting News, Date: 2023-11-09 . ...
Bears vs. Panthers final score 1 2 3 4 F Panthers 7 3 0 3 13 Bears 3 6 7 0 16. |
| 8 | 0.433 | 1 | Level 1: 347 | The Chicago Bears secured a narrow 16-13 victory over the Carolina Panthers on Thursday Night Football, driven ... The NFC remains highly competitive, and the legalization of online sports betting in Vermont is anticipated to boost NFL fan engagement. |
| 9 | 0.432 | 3 | Level 1: 347
Level 2: 349
Level 3: 357 | In a 'Thursday Night Football' game, the Chicago Bears narrowly defeated the Carolina Panthers 16-13, , ... with teams like the Tampa Bay Buccaneers, Dallas Cowboys, and Philadelphia Eagles dealing with their own strategic and injury-related issues. |
| 10 | 0.421 | 4 | Level 1: 347
Level 2: 349
Level 3: 357 | Tyson Bagent, the undrafted rookie quarterback for the Chicago Bears, started against the Carolina Panthers ... The game underscored the competitive disparities within the NFL, with teams like the Tampa Bay Buccaneers performing notably better. |

Table A.7: Qualitative Results: Retrieved content from deep nodes for comparative analysis.

| Query | Do the TechCrunch article on software companies and the Hacker News article on The Epoch Times both report an increase in revenue related to payment and subscription models, respectively? |
|---|---|
| **GT Response** | Yes |
| **MemTree Response** | No |
| **Error Analysis** | **Evidence Buried in Lengthy Retrieved Context.** The retrieved contents did contain the necessary information to correctly answer the question, but the relevant details were not immediately apparent due to being buried within a long and complex context. |

**MemTree Retrieved Content**

| Index | Sim | Node Depth | Ancestor Nodes | Content |
|---|---|---|---|---|
| 1 | 0.532 | 4 | Level 1: 202
Level 2: 203
Level 3: 562 | Founders, are events useful? Source: TechCrunch, Date: 2023-10-13 For the first time, the conference hosted an industry ... off 9% of its workforce across departments. |
| 2 | 0.519 | 3 | Level 1: 429
Level 2: 431
Level 3: 432 | How the conspiracy-fueled Epoch Times went mainstream and made millions Source: Hacker News, Date: 2023-10-16 Another says: ... dvideos and dance performances, will result in the salvation of humankind as the end of the world nears. |
| 3 | 0.498 | 2 | Level 1: 429
Level 2: 430 | **Title:** How the conspiracy-fueled Epoch Times went mainstream and made millions Source: Hacker News, Date: 2023-10-16 Falun Gong — or Falun Dafa, ... misfortune of being subscribed to the Epoch Times without my consent," one reads. |
| 4 | 0.488 | 3 | Level 1: 429
Level 2: 431
Level 3: 433 | How the conspiracy-fueled Epoch Times went mainstream and made millions Source: Hacker News ... Falun Gong religious community And they needed to make money A lot of it "Ensure that the paper gains a foothold in ordinary society and turns profitable," Li said. |
| 5 | 0.482 | 5 | Level 1: 661
Level 2: 663
Level 3: 1833 | Google fakes an AI demo, Grand Theft Auto VI goes viral and Spotify cuts jobs Source: TechCrunch, Date: 2023-12-09 " ... for startups offering consumer trading services directly — or indirectly, for that matter. |
| 6 | 0.473 | 3 | Level 1: 47
Level 2: 48
Level 3: 109 | Sam Altman backs teens' startup, Google unveils the Pixel 8 and TikTok tests an ad-free tier Source: TechCrunch, ... where oligopoly dynamics and first-mover advantages are shaping up and the value of proprietary data |
| 7 | 0.470 | 4 | Level 1: 661
Level 2: 662
Level 3: 1078 | OpenAI hosts a dev day, TechCrunch reviews the M3 iMac and MacBook Pro, and Bumble gets a new CEO Source: TechCrunch, Date: 2023-11-11 ... years later, now that her venture firm is also a decade old. |
| 8 | 0.470 | 4 | Level 1: 40
Level 2: 424
Level 3: 426 | **Title:** Here's how Rainforest, a budding Stripe rival, aims to win over software companies, , ... its Series B round (TechCrunch previously covered Kafene here ) Fintech firm Revio boosts community bank growth with $2.5M funding SkyWatch acquires Droneinsurance.com AP Automation Fintech Stampli announces $61M round led by Blackstone. |
| 9 | 0.453 | 3 | Level 1: 661
Level 2: 663
Level 3: 1834 | Uber's third-quarter earnings report showcases a company that is profitable but faces growth challenges The company reported an 11% year-over-year revenue increase to ... companies like WeWork and EV startup Arrival. |
| 10 | 0.451 | 5 | Level 1: 661
Level 2: 663
Level 3: 1834 | Uber's Q3 numbers include impressive profitability gains, slower-than-expected growth Source: TechCrunch, Date: 2023-11-07 On an adjusted basis, Uber Freight lost ... guidance More when we get Lyft's numbers |

Table A.8: Failure Example Analysis: Evidence Buried in Lengthy Retrieved Context

| Query | What entity, discussed in articles from both **The Verge** and **Fortune**, was involved in implementing a system to prevent **liquidation** due to software issues, **took on losses to maintain another company's balance sheet**, and claimed to have acted legally in its business practices as a **customer**, **payment processor**, and **market maker**? |
|---|---|
| GT Response | Alameda Research |
| MemTree Response | FTX |
| Error Analysis | **Insufficient Evidence.** The error occurred because the responder agent failed to capture the key points about Alameda Research from the retrieved content. The essential information was not sufficiently retrieved and was therefore not effectively extracted. |

**MemTree Retrieved Content**

| Index | Sim | Node Depth | Ancestor Nodes | Content |
|---|---|---|---|---|
| 1 | 0.529 | 7 | Level 1: 462
Level 2: 464
Level 3: 473 | The recent court testimonies have shed light on the intricate and problematic relationship between FTX and Alameda Research ..., exacerbated by the entangled operations with Alameda Research and the misleading assurances given to investors and employees. |
| 2 | 0.515 | 7 | Level 1: 462
Level 2: 464
Level 3: 473 | The recent court proceedings have highlighted significant misconduct and fraudulent practices within FTX and Alameda Research ... preferential treatment, which ultimately led to the collapse of FTX. |
| 3 | 0.510 | 6 | Level 1: 462
Level 2: 464
Level 3: 473 | The recent court proceedings have shed light on the intricate and fraudulent operations at FTX and Alameda Research. Gary Wang, co-founder and CTO, revealed that FTX misrepresented the amount in ... Bankman-Fried at the center of the fraudulent activities. |
| 4 | 0.503 | 4 | Level 1: 462
Level 2: 463
Level 3: 475 | Sam Bankman-Fried was a terrible boyfriend Source: The Verge, Date: 2023-10-10 Bankman-Fried , ... in bubbles along the main text Ellison wrote she was worried about "both actual leverage and presenting on our balance sheet." Bankman-Fried responded with a note: "Yup, and could also get worse." |
| 5 | 0.502 | 5 | Level 1: 40
Level 2: 42
Level 3: 43 | Sam Bankman-Fried, former CEO of FTX, testified about his limited knowledge and involvement in the financial operations ... at the financial mismanagement and liabilities, highlighting his insufficient oversight that led to the companies' problems. |
| 6 | 0.499 | 6 | Level 1: 40
Level 2: 42
Level 3: 43 | The jury finally hears from Sam Bankman-Fried Source: The Verge, Date: 2023-10-28 Risk is an inherent part of a futures exchange... Alameda Research's liabilities had gotten too high, and FTX was spending too much money on marketing. |
| 7 | 0.496 | 8 | Level 1: 462
Level 2: 464
Level 3: 473 | Today the FTX jury suffered through a code review Source: The Verge, Date: 2023-10-06 In the process of liquidating, FTX ... billion in customer money was gone, but on November 7th, Bankman-Fried tweeted "FTX is fine. Assets are fine." |
| 8 | 0.496 | 5 | Level 1: 462
Level 2: 464
Level 3: 474 | In the end, the FTX trial was about the friends screwed along the way Source: The Verge, Date: 2023-10-26 The defense ... liability that Ellison, Singh, Wang, and Bankman-Fried had known about in June 2022, we saw that $1.2 billion was a loan repayment to crypto lender Genesis. |
| 9 | 0.495 | 3 | Level 1: 462
Level 2: 463
Level 3: 475 | Sam Bankman-Fried and Caroline Ellison's intertwined personal and professional relationships significantly influenced the operations of FTX and Alameda Research ... eventual scrutiny of both entities. |
| 10 | 0.493 | 5 | Level 1: 462
Level 2: 464
Level 3: 939 | The trial of Sam Bankman-Fried, founder of the collapsed cryptocurrency exchange FTX, has exposed significant ... of the entanglements between FTX and Alameda Research, emphasizing the fraudulent nature of their operations. |

Table A.9: Failure Example Analysis: Insufficient Evidence

