# OpenReview forum: "From Isolated Conversations to Hierarchical Schemas: Dynamic Tree Memory Representation for LLMs"
_ICLR.cc/2025/Conference — ICLR 2025 Poster_

### Official Review · Reviewer_B6u5 · 2024-10-29

**Soundness:** 3
**Presentation:** 3
**Contribution:** 3
**Rating:** 6
**Confidence:** 4

**Summary:**

The paper proposes MemTree, an online algorithm for tackling the long-term memory issue of LLM inference. The approach, as shown in its name, builds a tree structure for managing chat history. The dynamic feature and hierarchical structure make MemTree effective and efficient.

**Strengths:**

The design of MemTree is reasonable and resembles the human cognition process in managing memory. Moreover, the retrieval mode of MemTree simply flattens the tree structure, making the retrieval more efficient than other traditional tree-based methods, and the dynamic feature of MemTree makes the approach work seamlessly with LLMs. Lastly, MemTree performs fairly well among the baseline methods.

**Weaknesses:**

- **More baseline models**: The paper only experiments with two models: GPT-4o and LLaMA2. As the design of MemTree is related to LLM (e.g., updating the parent nodes needs LLM-based operations), more baseline models are needed to better demonstrate the effectiveness of the proposed approach.

- **More details in efficiency comparison**: the paper claims that the proposed method closes the gaps between online and offline memory methods. What are the efficiencies of all the methods mentioned in the paper, including the naive one? How long is the inference time per instance?

- **Cost of using GPT-4o**: As the experiment tables are mostly about using GPT-4o, the cost of GPT-4o should also be considered as one aspect of the efficiency probation.

-**More human study**: More human study on comparing MemTree with different methods are needed, especially for ``human annotated evidence``. This could help demonstrate why MemTree is better and how it could be further improved.

**Questions:**

- **What is the reason for using GPT-4o as your baseline model as there are some other long-context LLMs, like long-lora, LongLLaMA, etc.?**

- **How is the MemTree compared to the prompt-compression methods, which aim at discarding the noisy information in the long contexts? For example, [1]**

[1] [LongLLMLingua: Accelerating and Enhancing LLMs in Long Context Scenarios via Prompt Compression](https://aclanthology.org/2024.acl-long.91) (Jiang et al., ACL 2024)

---

> ### Author Response · Authors · 2024-11-22
> **Response to Reviewer B6u5 (1/2)**
>
> > 1. More baseline models: The paper only experiments with two models: GPT-4o and LLaMA2 ... more baseline models are needed to better demonstrate the effectiveness of the proposed approach.
>
> Thank you for your comment. To clarify, our experiments covered a range of models, including two widely adopted LLMs (`GPT-4o` and `Llama-3.1-70B-Instruct`) and two embedding models (`text-embedding-3-large` and `E5-Mistral-7B-Instruct`). Additionally, we benchmark MemTree against five baselines: (1) a naive concatenation approach, (2) two online methods (*MemoryStream* and *MemGPT*), and (3) two offline methods (*RAPTOR* and *GraphRAG*). These baselines provide a comprehensive evaluation of MemTree's performance across both online and offline memory paradigms.
>
> In response to your feedback, we have extended our experiments to include smaller LLMs (`GPT-4o-mini` and `Llama-3.1-8B-Instruct`) and a smaller embedding model (`text-embedding-3-small`). As detailed in Section A.6.6 and available [here](https://sites.google.com/view/memtree-iclr2025/home#h.wxpugzne9r2v), the new Multihop RAG experiments confirm that MemTree's structured memory and accuracy remain robust and comparable even when using these smaller models.
>
> > 2. More details in efficiency comparison: ... the proposed method closes the gaps between online and offline memory methods. What are the efficiencies of all the methods mentioned in the paper, including the naive one?
>
> Thank you for your comment. In Section 4.2 (Baselines), we provide a detailed overview of the baseline methods used in our comparisons. It would be helpful to clarify which specific aspects of "efficiencies" you are referring to. To summarize that section, the baselines are categorized as follows:
>
> 1. Naive Approach: This method concatenates all contexts and feeds them directly to the LLM without utilizing external memory. While simple, it is highly resource-intensive and scales poorly for long-context scenarios due to the exponential growth in input size and processing costs.
>
> 2. Online Methods:
>    - MemoryStream and MemGPT allow dynamic updates to the memory representation without requiring a complete rebuild. These methods are efficient for frequent updates.
>    - However, they rely on tabular memory structures, which lack the hierarchical organization necessary to capture relationships between pieces of information.
>    - In contrast, MemTree introduces a structured, hierarchical memory representation that models these relationships more effectively.
>
> 3. Offline Methods:
>    - RAPTOR and GraphRAG require rebuilding the memory representation from scratch to incorporate new information, leading to significant computational overhead.
>    - These methods produce structured memory representations: RAPTOR uses hierarchical clustering with a fixed depth, while GraphRAG employs graph-based indexing.
>    - However, this static approach to memory updating is less efficient than online methods like MemTree, which dynamically integrates new information with far fewer operations.
>
> > 3. More details in efficiency comparison: ... How long is the inference time per instance?
>
> Thank you for your question. During inference, all models retrieve a fixed number of top-$K$ candidates: $K=10$ for MSC-E, QuALITY, and Multihop RAG, and $K=3$ for MSC. This standardization ensures consistency in retrieval performance and inference times across methods.
>
> MemTree, and the baselines: RAPTOR, MemGPT, and MemoryStream share a similar retrieval time complexity, as they all utilize a collapsed retrieval strategy. This approach treats memory instances as a flat set and ranks them by similarity to the query, ensuring efficient retrieval. In contrast, GraphRAG’s graph-based retrieval introduces additional overhead, as it requires traversing the graph structure to identify and extract relevant entities, making its inference time comparatively longer.

---

> ### Author Response · Authors · 2024-11-22
> **Response to Reviewer B6u5 (2/2)**
>
> > 4. More human study: More human study on MemTree ... This could help demonstrate why MemTree is better and how it could be further improved.
>
> Thank you for pointing this out. To address this, we designed a human study to evaluate how well the structure of MemTree aligns with human reasoning. Please refer to Section A.6.8 for details (also available [here](https://sites.google.com/view/memtree-iclr2025/home#h.gy8e88om3xmo)).
>
> To summarize, in this new experiment, we randomly selected a node in the tree as the source information and presented annotators with two alternative nodes: one a descendant of the source and another from a different subtree. Both alternatives were sampled at the same depth, uniformly distributed across the tree's depth. Annotators were asked to choose the alternative that they felt most closely matched the source information.
>
> This setup tests whether the hierarchical structure of MemTree aligns with human preferences. Our results showed that human preference matched the structure of MemTree with an accuracy of 93.2%. Additionally, we observed that this alignment improved as alternatives were sampled from deeper nodes, which contain more detailed information (92.05% at depths 2–5, 94.6% at depths 6–8, and 95.15% at depths 9–13). These findings demonstrate that MemTree’s structure effectively captures relationships that aligns with human intuition.
>
> > 5. Questions: What is the reason for using GPT-4o as your baseline model as there are some other long-context LLMs, like long-lora, LongLLaMA, etc.?
>
> Thank you for pointing this out. We selected `GPT-4o` and `Llama-3.1-70B-Instruct` (along with smaller models included in the revision, such as `GPT-4o-mini` and `Llama-3.1-8B-Instruct`) because they are widely adopted in both academic research and industry applications. This choice ensures that our results are directly comparable to existing methods, many of which are based on the same family of LLM models.
>
> For example, the baselines we compare against—GraphRAG (`GPT-4-turbo`), MemGPT (`GPT-4`, `GPT-4-turbo`), and RAPTOR (`GPT-3.5-turbo`)—rely on similar families of models. Using these models allows for a fair and consistent comparison across benchmarks.
>
> We have also updated the introduction to acknowledge the long-context LLMs you mentioned (e.g., long-lora, LongLLaMA).
>
> > 6. How is the MemTree compared to the prompt-compression methods, which aim at discarding the noisy information in the long contexts?
>
> Thank you for this suggestion. We believe the benefits of prompt compression and external memory methods, such as MemTree, are orthogonal. Prompt compression techniques like LLMLingua can be integrated on top of the retrieved content from any memory module, including MemTree.
>
> To address this, we conducted additional experiments to evaluate the impact of LLMLingua prompt compression on the MSC-E and Multihop RAG experiments (see Section A.6.9, also available [here](https://sites.google.com/view/memtree-iclr2025/home#h.e5itqlbwuqdb)). Our findings show that stricter compression rates (e.g., `rate=0.1`) significantly compromise retrieval task accuracy, while relaxed rates (closer to `rate=1`) restore accuracy to levels comparable to uncompressed content.
>
> Importantly, we also compared MemTree with another online method, MemoryStream, under varying compression rates. Across both experiments, MemTree consistently achieved higher accuracy, even under aggressive compression. This demonstrates that MemTree's robust structure preserves essential information more effectively, making it a superior complement to prompt compression methods.

---

> > ### Comment · Reviewer_B6u5 · 2024-11-26
> > **Response to author comments**
> >
> > Thanks for replying. My concerns have been resolved. The ratings have been updated.

---

> > > ### Author Response · Authors · 2024-11-27
> > > **Thank you for your feedback.**
> > >
> > > We’re glad to hear that our rebuttal resolved your concerns. Thank you for your follow-up and for updating your ratings. We truly appreciate your time and thoughtful review.

---

### Official Review · Reviewer_1Pfj · 2024-11-04

**Soundness:** 3
**Presentation:** 3
**Contribution:** 3
**Rating:** 6
**Confidence:** 4

**Summary:**

This paper proposes a knowledge organization method called MemTree for long-term memory storage. The main idea of MemTree is to hierarchically store information through a tree structure, enabling hierarchical organization and dynamic updating of memory.

The MemTree construction process resembles the concept of B-Trees in traditional computer science. First, the information to be stored is encoded into a vector. Starting from the root node of the tree, the most similar child node is selected based on LLM embedding similarity. If the similarity between the stored information and the closest child node exceeds a predeﬁned threshold, the process continues downwards until a leaf node is reached. Otherwise, the leaf node is expanded, and the information of the new node is summarized and propagated to its parent node. Experimental results conﬁrm that this method eﬀectively organizes memory content, which is beneﬁcial for tasks requiring knowledge retrieval.

**Strengths:**

Strengths:

1. The use of a tree structure eﬀectively organizes memory content, mimicking how humans summarize and store knowledge in higher-level memory constructs.

2. Experiments on MSC and MSC-E demonstrate the method’s strong and stable performance in longcontext memory tasks, while results on QuALITY show that MemTree can match or even surpass oﬄine methods (e.g., GraphRAG) with better update eﬃciency.

3. Compared to oﬄine methods such as GraphRAG and RAPTOR, MemTree oﬀers higher update eﬃciency while achieving comparable performance.

**Weaknesses:**

1. Collapsed Tree Retrieval for Knowledge Extraction: The paper adopts a Collapsed Tree Retrieval approach, presumably to retrieve not only leaf nodes but also aggregated, summary-based non-leaf nodes. However, this raises two questions:

Although non-leaf nodes summarize all descendant knowledge, this summarization could lead to information loss. How is this issue addressed?

If required knowledge spans nodes under diﬀerent parent nodes, would this tree structure still support retrieval of the complete set of needed information?

2. Tree Balance: While statistical analysis shows that MemTree remains generally balanced, there appears to be no explicit mechanism to enforce balance. And the observed balance may be due to the context not being suﬃciently long. More eﬀective operations for maintaining tree balance should be proposed, as the balance of the tree could theoretically impact the summarization quality of non-leaf nodes during MemTree construction.

3. Error Analysis for RAG Tasks: Could the paper provide a more detailed error analysis of MemTree’s performance on speciﬁc Retrieval-Augmented Generation (RAG) tasks? Speciﬁcally, is the source of errors due to inaccuracies in embedding similarity calculations, incomplete retrieved knowledge, or errors introduced by the summarization process?

4. Longer Context Retrieval: Could this method be extended to longer context retrieval tasks to further showcase its strengths in handling extensive knowledge storage and retrieval scenarios?

**Questions:**

see weaknesses

---

> ### Author Response · Authors · 2024-11-22
> **Response to Reviewer 1Pfj (1/2)**
>
> > 1. Collapsed Tree Retrieval: ... this summarization could lead to information loss. How is this issue addressed?
>
> We appreciate your feedback. During tree construction, content aggregation combines and incrementally abstracts node contents during insertion to create a hierarchical structure. Shallower nodes capture overarching themes and relationships between deeper nodes, while detailed information is retained within the deeper nodes themselves. This ensures that the hierarchy preserves both general and fine-grained information.
>
> In collapsed retrieval, the tree is flattened, and all nodes are treated as a single set for comparison. This approach ensures that detailed information from deeper nodes is fully accessible, even when higher-level aggregation is present. Queries with fine-grained details can still retrieve relevant content from deeper nodes, as they remain intact despite the aggregation at higher levels. This design ensures that MemTree’s hierarchical organization does not compromise the availability of detailed information during retrieval and addresses concerns of potential information loss.
>
> > 2. Collapsed Tree Retrieval: If required knowledge spans nodes under diﬀerent parent nodes, would this tree structure still support retrieval of the complete set of needed information?
>
> Thank you for your comment. Yes, the collapsed retrieval method addresses this concern by flattening the tree and treating all nodes as a single set for comparison. This ensures comprehensive coverage, allowing the retrieval of information spanning multiple parent nodes without being constrained by the tree hierarchy.
>
> In additional experiments, we compared collapsed retrieval with traversal retrieval to evaluate their effectiveness (see Section A.6.5, also available [here](https://sites.google.com/view/memtree-iclr2025/home#h.f7zzksh158c)). Collapsed retrieval achieves high accuracy by evaluating all nodes simultaneously and avoids the overhead of explicit traversal. In contrast, traversal retrieval iteratively selects top-$k$ nodes at each level, narrowing the search to localized paths. While traversal retrieval can be efficient with appropriately chosen $k$, it risks excluding relevant nodes at lower $k$ values ($k \leq 5$), leading to significant accuracy drops, particularly for complex queries.
>
> Our findings confirm that collapsed retrieval is better suited for scenarios where comprehensive evaluation is necessary, as it avoids hierarchical constraints while preserving structural characteristics through the content aggregation process during node construction. This makes collapsed retrieval particularly effective for queries requiring information from nodes under different parent nodes.

---

> ### Author Response · Authors · 2024-11-22
> **Response to Reviewer 1Pfj (2/2)**
>
> > 3. Tree Balance: While statistical analysis shows that MemTree remains generally balanced, there appears to be no explicit mechanism to enforce balance ... More eﬀective operations for maintaining tree balance should be proposed, as the balance of the tree could theoretically impact the summarization quality of non-leaf nodes during MemTree construction.
>
> Thank you for highlighting this concern. While MemTree does not employ a traditional balancing algorithm, its adaptive similarity threshold $\theta(d) = \theta_0 e^{\lambda d}$ inherently promotes balance during node insertion. This mechanism ensures that new nodes are added deeper into the tree only if they closely align with existing nodes at that depth; otherwise, they attach higher up, preventing disproportionately long branches and naturally distributing nodes.
>
> To address this further, we have added Appendix B (also available [here](https://sites.google.com/view/memtree-iclr2025/home#h.r226wz8beo20)), which provides a more thorough theoretical justification of MemTree by connecting it to online hierarchical clustering algorithms, particularly the Online Top-Down Clustering (OTD) method. As now outlined in Theorem 1 in the paper, MemTree achieves a hierarchy that is provably near-optimal under specific conditions, offering additional theoretical support for its balanced structure.
>
> While explicit balancing mechanisms could further optimize the tree, as you pointed our empirical analysis indicates that the current approach maintains a well-balanced tree and ensures effective summarization quality.
>
> > 4. Error Analysis for RAG Tasks: provide a more detailed error analysis of MemTree’s performance on speciﬁc RAG tasks? ... is the source of errors due to inaccuracies in embedding similarity calculations, incomplete retrieved knowledge, or errors introduced by the summarization process?
>
> Thank you for pointing this out. We agree that a detailed error analysis is an important addition. We are actively analyzing errors in MemTree's performance on the Multihop RAG experiments. While the analysis is ongoing, we have identified some preliminary insights:
>
> 1. Buried Evidence: In some cases, the evidence is present in the retrieved content but buried within lengthy text. This causes the LLM to overlook it when generating a response.
>
> 2. Insufficient Retrieval: Retrieving the top $K=10$ results occasionally fails to provide enough evidence to answer complex multihop queries requiring integration across multiple sources.
>
> 3. Bias in Yes/No Responses: For Yes/No queries, the LLM shows a bias toward positive responses (e.g., "Yes") even when the evidence is inconclusive or contradictory.
>
> We have shared our current findings [here](https://sites.google.com/view/memtree-iclr2025/home#h.wp5byt53g71q) to provide transparency during the discussion period. We aim to complete a comprehensive analysis and include the final results in the appendix of the revised manuscript before the deadline.
>
> > 5. Longer Context Retrieval: Could this method be extended to longer context retrieval tasks to further showcase its strengths in handling extensive knowledge storage and retrieval scenarios?
>
> Thank you for your input. The three experimental setups evaluated in this work—multi-session dialogues (MSC and MSC-E), single-document question answering (QuALITY), and multi-document question answering (Multihop RAG)—were specifically chosen to benchmark performance on long-context retrieval tasks.
>
> For instance, the Multihop RAG dataset requires queries to gather evidence from 609 articles, each averaging approximately 2,045 tokens (see Table A.1 for dataset statistics). This setup rigorously tests the memory's ability to efficiently retrieve and integrate information across extensive knowledge sources, showcasing MemTree’s strengths in handling long-context retrieval. By design, MemTree can scale to arbitrarily large knowledge storage, a property shared with all evaluated baselines.

---

> ### Comment · Reviewer_1Pfj · 2024-11-26
> **after response**
>
> Thanks for the detailed response, which solved most of my concerns. I think my score has already reflected my positive attitude, so I will maintain the score.

---

> > ### Author Response · Authors · 2024-11-27
> > **Thank you for your feedback.**
> >
> > Thank you for your response and for acknowledging that our rebuttal addressed your concerns. We truly value your time and thoughtful review.

---

### Official Review · Reviewer_1h4w · 2024-11-04

**Soundness:** 3
**Presentation:** 2
**Contribution:** 3
**Rating:** 6
**Confidence:** 3

**Summary:**

In this paper, the authors aim to address the challenge of long-term memory management in LLMs. Inspired by human cognitive patterns, they propose an algorithm named MemTree, which organizes information through a dynamic tree structure.

The authors evaluate the algorithm on conversational and document question-answering tasks and compare it with both online and offline knowledge representation methods. Their experimental results demonstrate that MemTree outperforms the various baselines presented.

**Strengths:**

- The paper includes extensive experiments and analysis.

- The performance of the proposed method surpasses most baselines.

**Weaknesses:**

- The paper frequently claims the efficiency of the proposed method but lacks specific experiments comparing the update and retrieval times with those of the baselines.

- The paper lacks details on how the memory is constructed for these evaluation datasets. Lines 303–306 indicate that the primary difference between MemTree and RAPTOR is that MemTree operates as an online algorithm, dynamically updating the tree memory representation with incoming knowledge, while RAPTOR applies hierarchical clustering on a fixed dataset. If this is the case, does it imply that MemTree’s memory size is typically smaller than RAPTOR’s during evaluation?

**Questions:**

- It is unclear why the approach in Section 3.2, which flattens the hierarchical structure, ensures more efficient retrieval. How does the mentioned retrieval process differ from directly traversing the entire memory to search for the closest node?

- It would be beneficial to conduct experiments demonstrating the impact of varying similarity thresholds, as this value influences the width of the tree.

- It is unclear what aspect of MemTree resembles human cognitive schemas—the dynamic memory updates or the tree structure? Human cognitive schemas appear to align more closely with a graph-like structure rather than a tree.

---

> ### Author Response · Authors · 2024-11-22
> **Response to Reviewer 1h4w (1/2)**
>
> > 1. Efficiency: ... the paper lacks specific experiments comparing the update and retrieval times with those of the baselines.
>
> Thank you for pointing this out. We have added new experiments comparing MemTree's update and retrieval times with those of RAPTOR, GraphRAG, detailed in Figure 5 (also available [here](https://sites.google.com/view/memtree-iclr2025/home#h.xlr2uwivdp4j)).
>
> - Update Efficiency: MemTree employs a concurrent update mechanism that parallelizes parent node updates on the CPU, enabling fast and scalable updates during conversations. As shown in Figure 5, MemTree inserts new information in approximately 10 seconds on average, compared to over an hour for RAPTOR and GraphRAG, which require costly offline memory tree reconstruction. While MemTree's cumulative time cost is 1.4x higher due to continuous updates, this trade-off allows real-time memory updates, making it well-suited for online scenarios.
>
> - Retrieval Efficiency: MemTree, RAPTOR, MemGPT, and MemoryStream utilize a collapsed retrieval strategy, ranking all memory instances as a flat set by query similarity. This ensures efficient retrieval, comparable across these methods. In contrast, GraphRAG introduces additional overhead due to its graph traversal requirements, resulting in slower inference times.
>
> > 2. Experiment details: ... the paper lacks details on how the memory is constructed for these evaluation datasets.
>
> Thank you for bringing this up. In Section 4.3, we summarize our experimental procedures, and we have provided additional details about how memory is constructed for each evaluation dataset in Appendix A.2.1. To address your concern, we have revised Appendix A.2.1 to ensure that all necessary details are now included.
>
> > 3. Experiment details: ... the primary difference between MemTree and RAPTOR is that MemTree operates as an online algorithm, dynamically updating the tree memory representation with incoming knowledge, while RAPTOR applies hierarchical clustering on a fixed dataset ... does it imply that MemTree’s memory size is typically smaller than RAPTOR’s during evaluation?
>
> Thank you for your insightful question. MemTree and RAPTOR differ fundamentally in their approaches. MemTree is a top-down, online algorithm that dynamically updates its tree structure as new information arrives, making it suitable for real-time applications. In contrast, RAPTOR is a bottom-up, offline method that constructs a hierarchical tree of fixed depth by clustering a fixed dataset.
>
> Regarding memory size during evaluation, MemTree's memory size is proportional to the amount of knowledge it has processed up to that point. This dynamic nature allows it to maintain a smaller memory footprint in scenarios with incremental data. In contrast, RAPTOR requires storing the entire fixed dataset along with its hierarchical structure, resulting in a larger memory footprint during evaluation, especially for large datasets.
>
> > 4. Retrieval: It is unclear why flat retrival ensures more efficient retrieval ... How does the mentioned retrieval process differ from directly traversing the entire memory to search for the closest node?
>
> Thank you for raising this point. To address your concern, we conducted additional experiments to clarify the differences in efficiency between collapsed retrieval and directly traversing the memory tree (see Section A.6.5, also available [here](https://sites.google.com/view/memtree-iclr2025/home#h.f7zzksh158c)).
>
> Collapsed retrieval flattens the tree, treating all nodes as a single set for comparison. This approach ensures comprehensive coverage by evaluating all nodes simultaneously, achieving high accuracy while avoiding the overhead of explicit traversal. In contrast, traversal retrieval utilizes the tree hierarchy by iteratively selecting top-$k$ nodes at each level, narrowing the search to localized paths. While traversal retrieval can be efficient with appropriately chosen $k$, it risks excluding relevant nodes at lower $k$ values ($k \leq 5$), leading to significant accuracy drops. Additionally, traversal retrieval may over-retrieve parent nodes, overlooking finer-grained details in deeper nodes, particularly in complex queries like temporal reasoning.
>
> Directly traversing the entire tree to find the closest node would combine the inefficiencies of both approaches, requiring evaluation of all nodes while maintaining hierarchical constraints. Collapsed retrieval, by comparison, simplifies computation by discarding hierarchy during evaluation. Importantly, the structural characteristics of MemTree are still implicitly preserved due to the content aggregation process during node construction.

---

> ### Author Response · Authors · 2024-11-22
> **Response to Reviewer 1h4w (2/2)**
>
> > 5. Adaptive threshold: It would be beneficial to conduct experiments demonstrating the impact of varying similarity thresholds, as this value influences the width of the tree.
>
> Thank you for this suggestion. We have conducted an ablation study to analyze the impact of varying the similarity threshold parameters on both the structure of the tree and downstream performance, as detailed in Section A.6.7 (also available [here](https://sites.google.com/view/memtree-iclr2025/home#h.kv41zftlarrt)).
>
> MemTree employs an adaptive similarity threshold $\theta(d) = \theta_0 e^{\lambda d}$, where $\theta_0$ controls the base similarity, and $\lambda$ determines how similarity threshold increase with depth. Higher base thresholds (e.g., $\theta_0 = 0.8$) result in shallower trees with more horizontal expansion, as stricter similarity criteria group nodes at higher levels. Conversely, lower base thresholds (e.g., $\theta_0 = 0.1$) encourage vertical growth, producing deeper trees where new nodes integrate at greater depths. While $\theta_0$ significantly impacts the tree's structure, varying $\lambda$ has a subtler effect, slightly influencing the maximum depth and node distribution.
>
> Our experiments on the MultiHop RAG dataset demonstrate that while these parameters shape the tree structure, the downstream task accuracy is robust across a wide range of values. For example, trees constructed with $\theta_0 = 0.1$ and $\lambda = 0.25$ showed slightly higher accuracy, but these differences were not statistically significant. This indicates that MemTree maintains its performance consistency regardless of variations in the threshold parameters.
>
> > 6. Connection to human cognition: It is unclear what aspect of MemTree resembles human cognitive schemas—the dynamic memory updates or the tree structure? Human cognitive schemas appear to align more closely with a graph-like structure rather than a tree.
>
> Thank you for your insightful question. The connection between MemTree and human cognitive schemas is primarily conceptual, centering on the dynamic updating of memory structures. While human cognitive schemas can indeed be graph-like—providing a holistic overview of knowledge—the tree structure in MemTree offers a natural, top-down approach to organizing information. This structure facilitates efficient assimilation of new data, much like how the brain integrates new experiences within existing knowledge, despite its more interconnected nature.
>
> To further evaluate the alignment between MemTree’s structure and human reasoning, we conducted a human study (Section A.6.8, also available [here](https://sites.google.com/view/memtree-iclr2025/home#h.gy8e88om3xmo)). In this experiment, annotators were presented with a source node and two alternatives: one a descendant of the source and another from a different subtree, sampled uniformly across the tree's depth. Annotators consistently selected the descendant as most related, achieving a 93.2% alignment with MemTree’s structure. This alignment improved at greater depths, ranging from 92.05% at depths 2–5 to 95.15% at depths 9–13.

---

> > ### Comment · Reviewer_1h4w · 2024-11-26
> >
> > Thank you for your clarification. Based on your response, I have decided to slightly increase my score.

---

> > > ### Author Response · Authors · 2024-11-27
> > > **Thank you for your feedback.**
> > >
> > > Thank you for taking the time to review our rebuttal and response. We are pleased to hear that it addressed your concerns and sincerely appreciate your valuable suggestions and effort in reevaluating our work.

---

### Official Review · Reviewer_Vzrr · 2024-11-06

**Soundness:** 2
**Presentation:** 3
**Contribution:** 2
**Rating:** 6
**Confidence:** 3

**Summary:**

This work proposes MemTree, a dynamic tree-structured memory representation algorithm designed for managing the storage, updating, and retrieval of external information. The authors' motivation is that existing memory management methods do not consider the intrinsic structure of external information, with individual information units being stored independently. The core of this work lies in the tree-based memory updating process, which involves tree traversal, leaf node expansion, and parent node aggregation updates. The work is validated across four benchmark datasets with different characteristics, showing particularly notable results in long-term dialogue scenarios.

**Strengths:**

S1. Clear Algorithm Characteristics. The work presents a clear and well-defined algorithm without making excessive claims. Both the algorithm description and the experimental design are straightforward and easy to follow.

S2. Experimental results effectively support the Authors' Claims.

S3. The primary contribution of this work lies in developing a tree-structured memory representation algorithm, with a focus on the processes of tree construction (node expansion) and updating.

**Weaknesses:**

W1. From another perspective, the generalizability of this work is limited, as its performance improvements are not as pronounced on non-long-term dialogue data.

W2. Although the work does not make excessive claims, its retrieval processes adopt methods from prior work. The retrieval process prioritizes performance by flattening the memory tree, which overlooks the structural characteristics of the information stored.

W3. Does the aggregation process rely heavily on the performance of the LLM? The paper does not provide sufficient discussion on this aspect. Since a substantial amount of information is abstracted and summarized during tree construction, it raises the question of whether semantic similarity-based matching remains accurate in this context.

**Questions:**

Minor Comment:
There is a typo in line 154: the first $C_v$ should be  $c_v$.

---

> ### Author Response · Authors · 2024-11-22
> **Response to Reviewer Vzrr (1/2)**
>
> > 1. Experiments: ... the generalizability of this work is limited, as its performance improvements are not as pronounced on non-long-term dialogue data.
>
> We appreciate your feedback. We summarize MemTree's performance on non-conversational tasks, as demonstrated on single-document and multi-document question answering benchmarks:
>
> * Single-document QA (*QuALITY*): MemTree outperforms MemoryStream, another online method, and surpasses RAPTOR, an offline baseline. It also achieves accuracy comparable (within 2%) to GraphRAG, demonstrating its effectiveness even on tasks with shorter contexts and no dialogue component (see Table 3).
>
> * Multi-document QA (*Multihop RAG*): MemTree handles complex reasoning tasks effectively, outperforming MemoryStream by a significant margin. Despite operating in an online setting, MemTree achieves accuracy within 0.5% of RAPTOR, an offline method with access to all information. Furthermore, MemTree surpasses all baselines, including RAPTOR and GraphRAG, on challenging temporal queries (see Table 4).
>
> Efficiency: MemTree is an online algorithm that supports continuous memory updates. This is a capability that offline methods such as RAPTOR and GraphRAG lack due to their reliance on accessing all available information and requiring memory reconstruction to incorporate new information. This efficiency, combined with MemTree’s competitive accuracy across non-dialogue tasks, highlights its broader applicability.
>
> > 2. Retrieval: Although the work does not make excessive claims, its retrieval processes adopt methods from prior work ... flattening the memory tree overlooks the structural characteristics of the information stored.
>
> Thank you for pointing this out. To address your concern, we conducted experiments comparing two retrieval strategies: collapsed retrieval and traversal retrieval (see Section A.6.5, also available [here](https://sites.google.com/view/memtree-iclr2025/home#h.f7zzksh158c)).
>
> Collapsed retrieval flattens the tree, treating all nodes as a single set for comparison. This approach ensures comprehensive coverage and achieves strong accuracy by evaluating all nodes simultaneously. It avoids the overhead of explicit traversal, simplifying computation. However, it disregards the tree's explicit hierarchical structure, relying instead on the implicit structure encoded during MemTree's construction through content aggregation.
>
> In contrast, traversal retrieval explicitly leverages the tree hierarchy by iteratively selecting top-$k$ nodes at each level, focusing on localized paths. While traversal retrieval preserves the explicit structure of the tree, it requires careful tuning of $k$ to balance accuracy and coverage. At higher $k$ values (e.g., $k=10$), traversal retrieval matches the accuracy of collapsed retrieval. However, at lower $k$ values ($k \leq 5$), its performance drops significantly due to prematurely narrowing the search space. Additionally, traversal retrieval can introduce redundancy by over-retrieving parent nodes, often missing finer-grained details in deeper nodes. This limitation is particularly evident in complex queries, such as those requiring temporal reasoning.
>
> Our findings highlight a trade-off between the two strategies. Collapsed retrieval simplifies computation and remains effective by leveraging MemTree’s implicit structure, while traversal retrieval explicitly aligns with the tree’s design but incurs additional overhead and requires careful parameter tuning to mitigate performance trade-offs.

---

> ### Author Response · Authors · 2024-11-22
> **Response to Reviewer Vzrr (2/2)**
>
> > 3. Aggregation: Does the aggregation process rely heavily on the performance of the LLM? ... Since a substantial amount of information is abstracted and summarized during tree construction, it raises the question of whether semantic similarity-based matching remains accurate in this context.
>
> Thank you for raising this important point. While MemTree’s aggregation process relies on LLM operations to abstract and summarize information (Section 3.1, Appendix A.1.2), this reliance is not unique; baselines like GraphRAG and RAPTOR also depend heavily on LLMs for tasks such as graph construction and hierarchical summarization. To further clarify, we conducted additional experiments to record the number of LLM calls required to build each memory baseline (Appendix A.3.4, also available [here](https://sites.google.com/view/memtree-iclr2025/home#h.t8vjm72mra1p)).
>
> To address concerns about the reliance on LLMs and embeddings performance for semantic similarity accuracy, new experiments with smaller LLMs and embedding models (Appendix A.3.6, also available [here](https://sites.google.com/view/memtree-iclr2025/home#h.wxpugzne9r2v)) demonstrate that MemTree retains strong performance even with resource-efficient models like `text-embedding-3-small` and `GPT-4o-mini`. Furthermore, the human study (Appendix A.3.8, also available [here](https://sites.google.com/view/memtree-iclr2025/home#h.gy8e88om3xmo)) shows a 93.2% alignment between MemTree’s structure and human preferences, validating that its summarization effectively preserves semantic relationships.

---

> > ### Author Response · Authors · 2024-11-27
> > **Follow-Up with Reviewer Vzrr**
> >
> > Thank you so much for taking the time to review our paper. We sincerely appreciate your constructive feedback and your positive evaluation of our work. We wanted to kindly follow up to see if there are any remaining concerns or questions that we can address.

---

> > > ### Comment · Reviewer_Vzrr · 2024-11-28
> > >
> > > The author's response addressed my concern and I decided to raise my rating.

---

### Author Response · Authors · 2024-11-22
**Response to All Reviewers - Summary of Changes**

We thank all reviewers for their thoughtful and constructive feedback. We are pleased that MemTree’s dynamic, tree-structured memory representation was recognized as an effective approach for long-term memory management in LLMs, with its ability to hierarchically organize and dynamically update memory content highlighted as a key strength [$\textcolor{purple}{\textrm{B6u5}}$, $\textcolor{navy}{\textrm{1Pfj}}$].

We appreciate the reviewers’ acknowledgment of MemTree’s clear and well-defined design, as well as the simplicity and rigor of our approach [$\textcolor{green}{\textrm{Vzrr}}$]. The efficiency of its retrieval mode, which flattens the tree structure for improved speed, was also noted [$\textcolor{purple}{\textrm{B6u5}}$]. We are glad that our experiments on benchmarks were viewed as strong validation of MemTree’s capabilities, with its performance in long-context memory tasks and comparisons to baselines like GraphRAG and RAPTOR being positively received [$\textcolor{teal}{\textrm{1h4w}}$, $\textcolor{navy}{\textrm{1Pfj}}$].

We believe that through your assistance, the paper has improved. The manuscript has been updated accordingly, and for convenient access, the revisions and additional results are available on this anonymous [website](https://sites.google.com/view/memtree-iclr2025/home).


1. **Theoretical Justification** [Section 3.1 and Appendix B] [$\textcolor{green}{\textrm{Vzrr}}$]: Strengthened the connection to online hierarchical clustering algorithms, providing a rigorous theoretical foundation for MemTree.

2. **Time Efficiency Comparison** [Figure 5] [$\textcolor{teal}{\textrm{1h4w}}$, $\textcolor{purple}{\textrm{B6u5}}$]: Updated Figure 5 to incorporate concurrent node updates and detailed comparisons of time costs with GraphRAG and RAPTOR.

3. **Memory Construction Details** [Appendix A.2.1] [$\textcolor{teal}{\textrm{1h4w}}$]: Revised the description of memory representation construction to clarify processes across all experiments.

4. **LLM Call Comparison** [Appendix A.3.4] [$\textcolor{teal}{\textrm{1h4w}}$]: Added experiments quantifying the number of LLM calls required by online and offline methods, highlighting computational costs.

5. **Retrieval Strategies Comparison** [Appendix A.3.5] [$\textcolor{green}{\textrm{Vzrr}}$, $\textcolor{teal}{\textrm{1h4w}}$]: Conducted new experiments comparing collapsed and traversal retrieval strategies, analyzing trade-offs in efficiency and accuracy.

6. **Robustness to Model Choices** [Appendix A.3.6] [$\textcolor{purple}{\textrm{B6u5}}$]: Added experiments using smaller LLMs and embedding models, demonstrating that MemTree retains strong performance even with resource-efficient setups.

7. **Adaptive Threshold Effects** [Appendix A.3.7] [$\textcolor{teal}{\textrm{1h4w}}$]: Investigated how adaptive threshold parameters affect tree structure and performance, providing insights into depth, breadth, and accuracy.

8. **Human Preference Alignment** [Appendix A.3.8] [$\textcolor{purple}{\textrm{B6u5}}$, $\textcolor{navy}{\textrm{1Pfj}}$]: Added a human evaluation experiment showing a strong alignment (93.2%) between MemTree’s structure and human preferences.

9. **Prompt Compression Impact** [Appendix A.3.9] [$\textcolor{purple}{\textrm{B6u5}}$]: Evaluated MemTree under varying prompt compression rates, demonstrating robustness and consistently superior performance over other online methods.

---

### Comment · Area_Chair_7T8S · 2024-11-25
**Action Required: Respond to Author Rebuttals - Nov 27**

Dear ICLR Reviewers,

The author discussion phase is ending soon. Please promptly review and respond to author rebuttals for your assigned papers. Your engagement is critical for the decision-making process.

Deadlines:
- November 26: Last day for reviewers to ask questions to authors.
- November 27: Last day for authors to respond to reviewers.
- November 28 - December 10: Reviewer and area chair discussion phase.

Thank you for your timely attention to this matter.

---

### Meta-Review · Area_Chair_7T8S · 2024-12-21

**Metareview:**

This work proposes MemTree, a dynamic tree-structured memory representation method for managing external information during LLM inference to enhance long context understanding. The approach builds a hierarchical tree structure for managing chat history, with its dynamic features and hierarchical organization contributing to both effectiveness and efficiency. The authors demonstrate MemTree's capabilities through evaluations on multi-turn dialogue understanding and document question answering benchmarks.

The authors have addressed most reviewer questions through extensive discussions, and reviewers have acknowledged the satisfactory responses. However, some aspects should be clarified in the final version, particularly the detailed methodology for memory structure construction and retrieval mechanisms.

Overall,I support acceptance at ICLR given the paper's thorough evaluation and demonstrated performance gains.

**Additional Comments On Reviewer Discussion:**

Minor issues/suggestions:
- "Hierachical" in the paper title should be corrected to "Hierarchical"
- Figure 1 could be removed as it adds limited value to the paper's presentation

---

### Decision · Program_Chairs · 2025-01-22

Accept (Poster)